# ORTHOSOLVER: A NEURAL PROPER ORTHOGONAL DECOMPOSITION SOLVER FOR PDES

**Ying Pang**[1,3,4]**, Jingyuan Wang**[1,2,3,]*** Jiahao Ji**[1,3]**, Fanhao Mu**[1,3]

[1]SKLCCSE, School of Computer Science and Engineering, Beihang University, Beijing, China
[2]School of Economics and Management, Beihang University, Beijing, China
[3]MOE Engineering Research Center of Advanced Computer Application Technology, Beihang University, Beijing, China
[4]National Superior College for Engineers, Beihang University, Beijing, China
{pangy, jywang, fanhaomu}@buaa.edu.cn, jiahaoji03@gmail.com

## ABSTRACT

Proper Orthogonal Decomposition (POD) is a cornerstone reduced-order modeling technique for accelerating the solution of partial differential equations (PDEs) by extracting energy-optimal orthogonal bases. However, POD's inherent linear assumption limits its expressive power for complex nonlinear dynamics, and its snapshot-based fixed bases generalize poorly to unseen scenarios. Meanwhile, emerging deep learning solvers have explored integrating decomposition architectures, yet their purely data-driven nature lacks essential physical priors and leads to modal collapse, where decomposed modes lose discriminative power. To address these challenges, we revisit POD from an information-theoretic perspective. We theoretically establish that POD's classical energy-maximization criterion is, in essence, a principle of maximizing mutual information. Guided by this insight, we propose OrthoSolver, a neural proper orthogonal decomposition framework that generalizes this core information-theoretic principle to the nonlinear domain. OrthoSolver iteratively and adaptively extracts a set of compact and expressive nonlinear basis modes by directly maximizing their mutual information with the data field. Furthermore, an orthogonality regularization is imposed to preserve the diversity of the learned modes and effectively mitigate mode collapse. Extensive experiments on seven PDE benchmarks demonstrate that OrthoSolver consistently outperforms state-of-the-art deep learning baselines.

## 1 INTRODUCTION

Partial Differential Equations (PDEs) constitute the fundamental language describing physical laws across numerous scientific and engineering disciplines (Wazwaz, 2002). However, high-fidelity numerical simulations of complex, real-world systems are often computationally prohibitive (Rozza et al., 2022). To address this issue, decomposition has emerged as a powerful paradigm, simplifying problem-solving by breaking down complex problems into a series of simpler, more tractable sub-tasks. This paradigm is central to the evolution of PDE-solving methodologies. Traditional numerical solvers, for instance, leverage Model Order Reduction (MOR) techniques, such as Proper Orthogonal Decomposition (POD), for acceleration (Carere et al., 2021). In parallel, data-driven models, spearheaded by deep learning, are evolving from monolithic architectures toward decompositional, multi-scale, and factorized structures to better model complex physical processes (Bhattacharya et al., 2021).

Within the domain of traditional MOR, Proper Orthogonal Decomposition (POD) is a cornerstone technique (Bright et al., 2013). Its core principle involves projecting a high-dimensional dynamical system onto a low-dimensional subspace spanned by a set of energy-optimal orthogonal basis functions, thereby achieving significant computational acceleration. Despite its widespread adoption, predicated on its mathematical optimality under linear assumptions, the efficacy of POD is

---
*Corresponding author

fundamentally constrained (Demo et al., 2023). First, it exhibits limited generalization, as basis functions generated for specific operating conditions often fail to extend to new scenarios. Second, POD's strict linear assumption precludes it from capturing the complex dynamics of highly non-linear systems. For example, when addressing multi-physics problems, POD typically decomposes each variable independently or assumes a linear correlation between them, thereby failing to capture the underlying nonlinear physical couplings (Lario et al., 2022).

Although deep learning-based PDE solvers have advanced rapidly, they face distinct bottlenecks. Monolithic operator architectures, such as the Fourier Neural Operator (FNO) (Li et al., 2021) and DeepONet (Lu et al., 2021), have limited modeling capacity when confronted with complex scenarios (Wu et al., 2023). In response, subsequent work has explored more sophisticated architectures; U-Net-like (Ronneberger et al., 2015) models including LSM (Wu et al., 2023), U-NO (Rahman et al., 2022), and U-FNO (Wen et al., 2022) employ multi-scale structures, whereas models like Transolver (Wu et al., 2024), Factformer (Li et al., 2023a), and F-FNO (Tran et al., 2023) utilize factorization and data slicing to simplify the problem. While these decompositional strategies like Transolver provide a powerful means of simplifying complexity, these approaches often lack effective mechanisms to enforce independence among the decomposed components, leading to mode collapse in complex scenarios—a phenomenon wherein the decomposed modes become indistinguishable, thus losing their differential representation power (Luo et al., 2025). Consequently, a critical challenge has emerged: designing a unified framework that combines efficient decomposition with robust nonlinear modeling while simultaneously preventing mode collapse.

The crux of this challenge stems from a fundamental dichotomy: while traditional decomposition methods like POD rest upon solid mathematical foundations, their reliance on variance-based metrics introduces significant errors when applied to nonlinear physical systems. Conversely, the decomposition strategies employed in existing data-driven models often lack a sound theoretical underpinning. To address this gap, this work revisits POD from an information-theoretic perspective, theoretically establishing that its core principle is equivalent to the maximization of Mutual Information (MI) under a linear-Gaussian assumption (Chechik et al., 2003; Burges et al., 2010). This theoretical link not only elucidates the foundations of POD but also underscores the limitations of variance-based approaches in nonlinear regimes. Consequently, MI, as a universal measure of statistical dependence unconstrained by linearity, provides a more principled framework for capturing the intricate correlations characteristic of nonlinear systems (Globerson & Tishby, 2003).

This insight motivates the proposed framework, OrthoSolver: a deep proper orthogonal decomposition framework guided by the principle of maximizing mutual information. OrthoSolver comprises two key modules: a orthogonal basis extraction module predicated on the maximum mutual information principle and a dynamics evolution module inspired by POD. The former leverages an information-theoretic objective to extract salient modes from the nonlinear dynamics, while the latter performs efficient evolution within the resulting low-dimensional space. To mitigate mode collapse, an orthogonality regularization is introduced to enforce the independence and representational efficacy of the decomposed modes. The contributions of this paper are as follows:

- Theoretical Contribution: We reveal and formally establish a deep theoretical connection between POD and mutual information maximization, proving that POD's energy-optimal orthogonal basis decomposition is a special case of maximum mutual information under the linear Gaussian assumption.

- Methodological Contribution: We propose OrthoSolver, an end-to-end, information-theoretic-guided deep learning framework for orthogonal decomposition that extends the physical principles of POD to the nonlinear domain.

- Extensive experiments on seven benchmark datasets, spanning a diverse range of physical phenomena across both 1D and 2D PDEs, demonstrate that OrthoSolver significantly outperforms existing state-of-the-art methods.

## 2 RELATED WORK

**Neural Operators for PDEs.** Learning solution operators for entire families of PDEs is a central theme in scientific machine learning. Dominant approaches include DeepONet (Lu et al., 2021) and the Fourier Neural Operator (FNO) (Li et al., 2021), which parameterizes global convolutions

efficiently using the FFT. The success of FNO has inspired a range of extensions and hybrid architectures. To better capture features at different resolutions, several works have combined FNO with the hierarchical U-Net (Ronneberger et al., 2015) architecture, leading to models like U-FNO (Wen et al., 2022) and U-NO (Rahman et al., 2022). Other extensions have adapted the core idea to handle complex geometries (Geo-FNO (Li et al., 2023b)) or leverage alternative transforms like wavelets (MWT (Gupta et al., 2021)). Concurrently, Transformers (Vaswani et al., 2017), a cornerstone of deep learning, have been successfully adapted for solving PDEs. Models like FactFormer (Li et al., 2023a) leverage low-rank structures to boost efficiency, while others like OFormer (Li et al., 2022) and GNOT (Hao et al., 2023) address the quadratic complexity of attention by incorporating linear Transformer variants. However, these monolithic operator learning approaches often struggle to generalize in complex physical scenarios, facing challenges with intricate boundary conditions and capturing the full spectrum of physical dynamics (Wu et al., 2023). Our work instead adopts a decompositional paradigm, aiming for greater interpretability and robustness by breaking down complex fields into simpler, fundamental components.

**Decomposition-based Models for PDEs.** An alternative paradigm focuses on decomposing the solution field. One line of work employs purely data-driven decompositions; for instance, LSM (Wu et al., 2023) maps solutions to a latent spectral basis, while Transolver (Wu et al., 2024) decomposes the input into learnable slices. A key challenge for these methods is the lack of physical grounding, which can lead to mode collapse, where learned basis functions become redundant (Luo et al., 2025). Another line of work leverages traditional physics-based decomposition, such as Proper Orthogonal Decomposition (POD), and then uses a neural network to evolve the mode coefficients (e.g., POD-DeepONet (Lu et al., 2022)). This hybrid approach, however, is fundamentally constrained by the linear decomposition error inherent in POD when applied to nonlinear systems. While some methods (Lario et al., 2022) attempt to learn this residual error with machine learning, they do not address the core limitation of the initial linear decomposition. Notably, several theoretical works have explored the connection between Proper Orthogonal Decomposition (POD) and mutual information from an information-theoretic perspective (Chechik et al., 2003; Globerson & Tishby, 2003; Burges et al., 2010). However, these studies remain confined to proving theoretical equivalences and do not attempt to leverage this insight to generalize POD. In contrast, our work builds directly on this theoretical foundation and presents the first framework that extends POD to the nonlinear regime. OrthoSolver achieves this by learning a nonlinear decomposition guided by a physically principled objective, thereby simultaneously circumventing mode collapse and the linear approximation error inherent in classical POD.

**Information Theory in Deep Learning.** The use of Mutual Information (MI) as an objective has proven to be a powerful tool for learning structured and disentangled representations in deep learning. Foundational concepts such as the InfoMax principle (Veličković et al., 2018) and the Information Bottleneck principle (Tishby et al., 2000) have been widely applied in representation learning to distill salient features. While direct MI computation is intractable, this challenge has spurred the development of a rich literature on neural MI estimators that provide tractable bounds, including MINE (Belghazi et al., 2018), InfoNCE (Oord et al., 2018), and CLUB (Cheng et al., 2020), among others. OrthoSolver innovatively bridges the gap between this line of research and scientific computing. To the best of our knowledge, this is the first framework to leverage these modern MI estimators to guide the decomposition of PDE solutions.

## 3 PRELIMINARIES

### 3.1 PROBLEM SETUP: OPERATOR LEARNING

In classical machine learning, the goal is to learn a mapping between finite-dimensional Euclidean spaces, $f : \mathbb{R}^{d_{in}} \rightarrow \mathbb{R}^{d_{out}}$, using a dataset of input–output pairs $\mathcal{P} = \{(\mathbf{x}_i, \mathbf{y}_i)\}_{i=1}^N$. While effective for many tasks, this vector-to-vector paradigm is not naturally suited for problems governed by PDEs, where the goal is to learn a mapping between *functions*.

Consider a parametric PDE of the form $(\mathcal{L}_a u)(x) = f(x)$, where $\mathcal{L}_a$ is a differential operator parameterized by a function $a(x)$ (encoding boundary conditions, coefficients, or source terms). For each $a(x)$, there exists a solution $u(x)$. Learning across this PDE family amounts to approximating the *solution operator* that maps input functions $a$ to output functions $u$. This motivates the operator-learning paradigm, which seeks to learn mappings between infinite-dimensional function spaces.

**Operator Learning** (Anandkumar et al., 2020) can be defined as:

$$\mathcal{F} : \mathcal{X} \times \Theta \to \mathcal{Y}, \tag{1}$$

where $\mathcal{X} = \{x \,|\, x : \Omega \to \mathbb{R}^{d_x}\}$ and $\mathcal{Y} = \{y \,|\, y : \Omega \to \mathbb{R}^{d_y}\}$ denote input and output function spaces over domain $\Omega \subset \mathbb{R}^d$, $d_x$ and $d_y$ are channel dimensions, and $\Theta$ denotes the parameters.

## 3.2 PROPER ORTHOGONAL DECOMPOSITION (POD)

Proper Orthogonal Decomposition (POD) (Rozza et al., 2022) is a classical model reduction technique that formulates **an explicit variance-maximization problem** (see Appendix C.1 for derivation). Its goal is to identify a low-dimensional subspace that captures as much data variability ("energy") as possible from the snapshots $\{u_i\}_{i=1}^N$. Treating each snapshot as a realization of a random variable $\mathbf{u} \in \mathbb{R}^m$, POD formally constructs the orthonormal basis $\{\phi_k\}_{k=1}^r$ by iteratively solving the following sequential optimization problem:

$$\phi_k = \underset{\substack{\|\phi\|=1 \\ \phi \perp \{\phi_1, \ldots, \phi_{k-1}\}}}{\arg\max} \quad \mathrm{Var}(\langle \mathbf{u}, \phi \rangle). \tag{2}$$

where the objective explicitly seeks directions that maximize the projected variance subject to orthogonality constraints.

Practically, this optimal basis is obtained via a linear matrix factorization, typically Singular Value Decomposition (SVD) of the snapshot matrix $\mathbf{U} = [u_1, \ldots, u_N]$. Any snapshot can then be approximated as a linear combination of these basis modes:

$$u_i \approx \sum_{k=1}^r a_{ik} \phi_k, \tag{3}$$

where $a_{ik} = \langle u_i, \phi_k \rangle$ is the projection coefficient. *By maximizing variance, POD finds the optimal linear subspace.* However, its reliance on linear factorization limits its capacity to capture complex nonlinear dynamics, motivating data-driven generalizations.

## 3.3 BEYOND LINEAR CORRELATION: MUTUAL INFORMATION

While variance maximization, used in POD, is sensitive to linear correlations, it does not capture more complex statistical structures. In contrast, Mutual Information (MI) from information theory provides a general measure of dependency (Globerson & Tishby, 2003). Defined as $I(X;Y) = H(X) - H(X|Y)$, MI quantifies the reduction in uncertainty about one variable given knowledge of another. Crucially, its formulation allows it to capture arbitrary nonlinear relationships, making it a more comprehensive tool for dependency analysis than linear correlation.

## 4 METHODOLOGY: THE ORTHOSOLVER FRAMEWORK

### 4.1 FROM VARIANCE TO MUTUAL INFORMATION: A PRINCIPLED GENERALIZATION

As Section 3.2 mentioned, the fundamental limitation of Proper Orthogonal Decomposition (POD) is its reliance on maximizing projected variance as equation 2, a principle optimal for linear systems but ill-suited for complex, nonlinear dynamics. To overcome this, we re-contextualize POD from an information-theoretic perspective, positing that its variance-based objective is a constrained special case of a more general principle **maximizing mutual information (MI)**. Existing theoretical works have formally proven that maximizing variance is equivalent to maximizing MI under linear Gaussian assumptions (Chechik et al., 2003; Globerson & Tishby, 2003; Burges et al., 2010):

**Theorem 1.** *Assume that the data snapshots $\mathbf{u}$ follow a multivariate Gaussian distribution and that the projection $a = \langle \mathbf{u}, \phi \rangle$ is a linear operation. Then, maximizing the projection variance $\mathrm{Var}(a)$ is equivalent to maximizing the mutual information $I(\mathbf{u}; a)$ between the original data and its projection coefficient.*

*Proof.* Since $\mathbf{u}$ is Gaussian, its linear projection $a$ is also a univariate Gaussian random variable. The differential entropy of a zero-mean Gaussian variable is given by

$$H(a) = \frac{1}{2} \log(2\pi e \cdot \mathrm{Var}(a)). \tag{4}$$

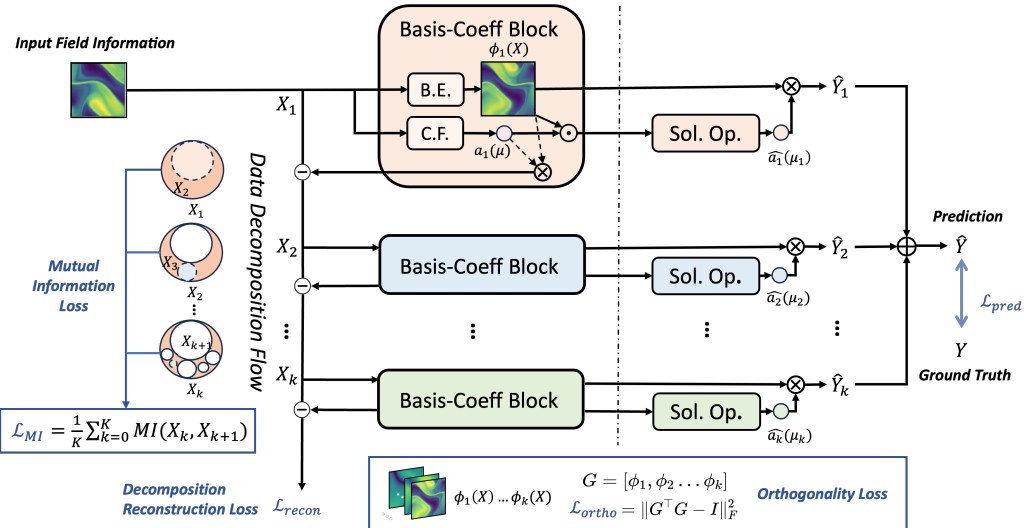

Figure 1: **Overall architecture of OrthoSolver.** The input function $\mathbf{u}(\boldsymbol{\mu})$ is first decomposed into global basis functions and coefficients, which are then evolved in the latent space by the solver and recombined to produce $\hat{\mathbf{Y}}(\boldsymbol{\mu}')$.

Because the logarithm is a monotonic function, maximizing the variance $\mathrm{Var}(a)$ is equivalent to maximizing the entropy $H(a)$:

$$\arg\max \mathrm{Var}(a) \iff \arg\max H(a). \tag{5}$$

The mutual information between $\mathbf{u}$ and $a$ is defined as $I(\mathbf{u}; a) = H(a) - H(a|\mathbf{u})$. Since $a$ is a deterministic function of $\mathbf{u}$, its conditional entropy vanishes, i.e., $H(a|\mathbf{u}) = 0$. Therefore, $I(\mathbf{u}; a) = H(a)$. Combining these results yields:

$$\arg\max \mathrm{Var}(a) \iff \arg\max H(a) \iff \arg\max I(\mathbf{u}; a). \tag{6}$$

This proves that under linear Gaussian conditions, the variance-maximization objective of POD is equivalent to a mutual-information-maximization objective. □

Existing theoretical studies have proven Theorem 1 that POD's variance maximization is equivalent to mutual information maximization under linear Gaussian assumptions. However, these works remain largely theoretical and have not leveraged this insight to practically generalize POD. We observe that Theorem 1 reveals a key insight: POD's use of variance to identify dominant bases is essentially a specific instance of using mutual information. However, the variance maximization criterion is fundamentally limited to capturing second-order moments, making it ill-suited for characterizing the complex, high-order dependency structures in highly nonlinear PDEs. By contrast, mutual information can effectively capture complex nonlinearities. Building on Theorem 1, we generalize the linear decomposition paradigm of POD from the restricted variance metric to the general mutual information metric. This leads to the development of OrthoSolver, which adaptively extracts the most informative bases in nonlinear spaces, thereby extending the core philosophy of POD to nonlinear decomposition.

## 4.2 ORTHOSOLVER: MODEL ARCHITECTURE

We generalize the classical POD framework to an operator-learning setting by leveraging information-theoretic principles. Unlike traditional POD, which relies on a fixed linear basis obtained from snapshot matrices, OrthoSolver adaptively learns a nonlinear basis that maximizes mutual information with the data, enabling more expressive and data-efficient representations.

**Neural POD Operator Learning.** In contrast to conventional operator learning methods that seek to learn a direct mapping $\mathcal{F} : \mathcal{X} \to \mathcal{Y}$ (Anandkumar et al., 2020), the proposed framework decomposes this process into a composition of three distinct operators:

$$\mathcal{F} = \mathcal{D} \circ \mathcal{S}_\theta \circ \mathcal{E}_\theta \tag{7}$$

where $\circ$ denotes operator composition. The **Basis Decomposition Operator** $\mathcal{E}_\theta : \mathcal{X} \to (\Phi, \mathbb{R}^K)$ maps an input function $x(\mu)$ to a set of $K$ global basis functions $\{\Phi_k\}$ and corresponding coefficients $\{a_k(\boldsymbol{\mu})\}$. The **Solver Operator** $\mathcal{S}_\theta$ evolves these coefficients to new parameter conditions $\boldsymbol{\mu}'$. Finally, the **Synthesis Operator** $\mathcal{D}$ reconstructs the solution by linear superposition: $\hat{\mathbf{Y}}(\boldsymbol{\mu}') = \sum_{k=1}^K \hat{a}_k(\boldsymbol{\mu}')\Phi_k$.

**Adaptive Basis Learning.** Guided by the principle established in Section 4.1, our decomposition operator $\mathcal{E}_\theta$ operationalizes the generalization of POD. Instead of maximizing variance, its core objective is to learn a set of basis functions $\{\Phi_k\}$ that maximize the mutual information (MI) :

$$\Phi_k^\star = \arg \max_{\{\Phi_k\}} I\big(\mathrm{Proj}(\mathbf{u}, \{\Phi_k\}); \mathbf{u}\big), \tag{8}$$

where $\mathrm{Proj}(\cdot)$ represents the projection of the data onto the learned basis. This information-theoretic objective drives the learning of a compact and expressive basis capable of capturing complex nonlinear dependencies. To implement this, we design $\mathcal{E}_\theta$ as a residual-based sequential process, where each step extracts the single most informative basis-coefficient pair $(\Phi_k, a_k(\boldsymbol{\mu}))$ from the current data field.

**Data Flow.** Figure 1 illustrates the data flow: (i) decomposition onto global basis, (ii) coefficient evolution in a low-dimensional latent space, and (iii) synthesis into the final solution. This modular factorization provides interpretability and allows efficient generalization to new parameter regimes.

## 4.3 BASIS DECOMPOSITION MODULE VIA MUTUAL INFORMATION MAXIMIZATION

Our basis decomposition module implements the information-theoretic principle outlined in Section 4.1. It mimics the sequential, residual-based process of POD but replaces the linear, variance-driven objective with a non-linear, information-driven one. The module iteratively extracts basis-coefficient pairs to greedily maximize the information captured from the data field.

Formally, the input function $\mathbf{u}(\boldsymbol{\mu})$ is defined as the initial residual $\mathbf{X}_1$. At each step $k$, this module produce a basis–coefficient pair $(\Phi_k, a_k)$ by solving

$$\max_{\Phi_k, a_k} I(\mathbf{X}_k, a_k), \tag{9}$$

where $I(\cdot, \cdot)$ denotes the mutual information between the residual and the extracted coefficient. Intuitively, this ensures that $\Phi_k$ represents the most informative mode contained in $\mathbf{X}_k$. The `BasisExtractor` is realized as a Factorized Fourier Neural Operators (F-FNO) (Tran et al., 2023), selected for its efficacy in operator learning tasks within function spaces, while the `CoeffExtractor` is a Multi-Layer Perceptron (MLP):

$$\Phi_k = \mathrm{FNO}(\mathbf{X}_k; \boldsymbol{\theta}_{\mathrm{fno},k}), \tag{10}$$

$$a_k = \mathrm{MLP}(\mathbf{X}_k; \boldsymbol{\theta}_{\mathrm{mlp},k}). \tag{11}$$

The residual is then updated as

$$\mathbf{X}_{k+1} = \mathbf{X}_k - a_k \Phi_k, \tag{12}$$

and the process is repeated for $K$ steps.

For practical optimization, direct maximization of $I(\mathbf{X}_k, a_k)$ is computationally challenging. We reformulate the objective as minimizing the mutual information between the current representation and the residual after extracting $a_k$, i.e.,

$$\min I(\mathbf{X}_k, \mathbf{X}_k - a_k \Phi_k) = \min I(\mathbf{X}_k, \mathbf{X}_{k+1}) \tag{13}$$

where $\mathbf{X}_{k+1}$ denotes the residual representation. As demonstrated in Appendix C.2, maximizing the information captured by the mode is equivalent to minimizing the information carried over in the residual. The final MI-based loss is formulated as the average over all steps:

$$\mathcal{L}_{\mathrm{mi}} = \frac{1}{K} \sum_{k=1}^K I(\mathbf{X}_k, \mathbf{X}_{k+1}) \tag{14}$$

**MI Estimation.** Direct computation of mutual information is generally intractable in deep learning, as it requires access to the true joint and marginal distributions. To address this, we adopt a variational approach that optimizes a tractable upper bound of the MI. Specifically, we employ the Contrastive Log-ratio Upper Bound (CLUB) (Cheng et al., 2020), which provides a tight and efficiently trainable surrogate. For a given pair $(\mathbf{X}_k, a_k)$, CLUB estimates the MI by learning a variational distribution $q(a_k|\mathbf{X}_k)$ to approximate the true posterior $p(a_k|\mathbf{X}_k)$. In practice, this conditional distribution can be parameterized by a neural network, such as a multi-layer perceptron (MLP). Then the upper bound can be caculated by:

$$I_{\text{CLUB}}(\mathbf{X}_k, a_k) = \mathbb{E}_{p(\mathbf{X}_k, a_k)}[\log q(a_k|\mathbf{X}_k)] - \mathbb{E}_{p(\mathbf{X}_k)p(a_k)}[\log q(a_k|\mathbf{X}_k)] \tag{15}$$

This estimator can be efficiently optimized using samples from the training batch. This tractable estimate is then substituted into Equation equation 14 to facilitate end-to-end training.

**Reconstruction Constraint.** To ensure the learned basis functions and coefficients can accurately reconstruct the original input data, a reconstruction constraint is imposed. This loss penalizes the discrepancy between the original function and its reconstruction from the full set of extracted modes:

$$\mathcal{L}_{\text{recon}} = \left\| \mathbf{u} - \sum_{k=1}^{K} a_k(\mathbf{u})\mathbf{\Phi}_k \right\|_F^2 \tag{16}$$

**Basis Orthogonality Constraint.** As noted in (Luo et al., 2025; Doimo et al., 2022), in complex dataset scenarios, deep learning models employing decomposition strategies such as (Wu et al., 2024) may suffer from mode collapse, a phenomenon where optimizers tend to converge towards the redundant features rather than decoupling truly independent components. Mathematically, this phenomenon manifests as high similarity between the learned basis vectors (i.e., $\phi_i \approx \phi_j$), leading to approximate linear dependence among the column vectors of the basis matrix $\mathbf{G}$. This results in a **decrease in the effective rank** of $\mathbf{G}$ (i.e., rank$(\mathbf{G}) < K$), thereby limiting the representational capacity of the subspace.

To theoretically strictly avoid this degeneracy and promote basis diversity, we introduce an orthogonality constraint. By regularizing the Gram matrix of the basis functions to approximate the identity matrix (i.e., $\mathbf{G}^T\mathbf{G} \approx \mathbf{I}$), we theoretically ensure the linear independence of the basis vectors, thereby maintaining the **full-rank property** of the decomposition (rank$(\mathbf{G}) \approx K$). The loss function is defined using the Frobenius norm as follows:

$$\mathcal{L}_{\text{ortho}} = \left\| \mathbf{G}^T\mathbf{G} - \mathbf{I} \right\|_F^2 \tag{17}$$

where $\mathbf{G} = [\mathbf{\Phi}_1, \mathbf{\Phi}_2, \ldots, \mathbf{\Phi}_K]$ is the matrix consisting of flattened basis vectors as columns, and $\mathbf{I}$ is the identity matrix.

## 4.4 Dynamics Evolution and Solution Synthesis

Once the decomposition module has extracted the global basis functions $\{\mathbf{\Phi}_k\}$ and their corresponding coefficients $\{a_k(\boldsymbol{\mu})\}$, the **Solver Operator** $\mathcal{S}_\theta$ predicts the system's evolution in the low-dimensional latent space.

For each mode $k$, a dedicated F-FNO-based `SolutionOperator` evolves the coefficient $a_k(\boldsymbol{\mu})$ to a new parameter state $\boldsymbol{\mu}'$. The operator takes the coefficient history and the static basis function as input to predict the new coefficient $\hat{a}_k(\boldsymbol{\mu}')$:

$$\hat{a}_k(\boldsymbol{\mu}') = \text{FNO}_k\left(\text{Concat}\left(a_k(\boldsymbol{\mu}), \mathbf{\Phi}_k\right)\right) \tag{18}$$

Once the set of predicted coefficients $\{\hat{a}_k(\boldsymbol{\mu}')\}_{k=1}^{K}$ has been inferred for all modes, the **Synthesis Operator** $\mathcal{D}$ assembles the final high-dimensional solution. This is a parameter-free linear combination of the global basis functions weighted by their newly predicted coefficients:

$$\hat{\mathbf{Y}}(\boldsymbol{\mu}') = \sum_{k=1}^{K} \hat{a}_k(\boldsymbol{\mu}')\mathbf{\Phi}_k \tag{19}$$

To train the solver networks, we define a prediction loss that measures the discrepancy between the synthesized solution and the ground truth. We use the Relative L2 error, which is common for evaluating physics-based learning problems:

$$\mathcal{L}_{\text{pred}} = \frac{\|\mathbf{Y}(\boldsymbol{\mu}') - \hat{\mathbf{Y}}(\boldsymbol{\mu}')\|_2}{\|\mathbf{Y}(\boldsymbol{\mu}')\|_2} \tag{20}$$

### 4.5 MODEL TRAINING

The OrthoSolver framework is trained end-to-end by minimizing a composite objective function that incorporates the four distinct loss terms derived in the previous sections. The overall loss, $\mathcal{L}_{\text{total}}$, is a dynamically weighted sum of these components, designed to balance the multifaceted goals of our model.

**Total Loss Function.** The total loss is composed of four terms, each targeting a specific aspect of the learning process. The Mutual Information loss ($\mathcal{L}_{\text{MI}}$) from Eq. equation 14 drives the extraction of informative basis functions. The Reconstruction loss ($\mathcal{L}_{\text{recon}}$) from Eq. equation 16 ensures the decomposition is faithful to the original data. The Orthogonality loss ($\mathcal{L}_{\text{ortho}}$) from Eq. equation 17 encourages diversity among the basis functions. Finally, the Prediction loss ($\mathcal{L}_{\text{pred}}$) from Eq. equation 20 trains the solver to accurately evolve the latent coefficients. The combined objective is:

$$\mathcal{L}_{\text{total}} = \lambda_{\text{MI}}\mathcal{L}_{\text{MI}} + \lambda_{\text{recon}}\mathcal{L}_{\text{recon}} + \lambda_{\text{ortho}}\mathcal{L}_{\text{ortho}} + \lambda_{\text{pred}}\mathcal{L}_{\text{pred}} \tag{21}$$

where $\lambda$ are the weights for each loss component. To balance these multi-task objectives, Dynamic Weight Averaging (DWA) is employed (Liu et al., 2019).

## 5 EXPERIMENTS

To comprehensively evaluate the performance and robustness of our proposed OrthoSolver framework, we conduct extensive experiments on a diverse suite of 7 benchmark datasets from the field of fluid dynamics. These datasets span both 1D and 2D problems and cover a range of physical phenomena and complexities. Reproducibility details like code and datasets can be obtained in section Reproducibility Statement. A comprehensive analysis of model efficiency, including parameters, training times, and memory consumption, is detailed in the Appendix D.6.

### 5.1 DATASETS AND BASELINES

Our evaluation is performed on seven benchmark datasets from PDEBench (Takamoto et al., 2022), detailed in Appendix D.1. They include canonical problems such as Burgers' equation and Advection equation, as well as more challenging simulations like time-dependent Navier-Stokes flow, allowing us to test the model's ability to handle varying levels of non-linearity and dimensionality.

We benchmark OrthoSolver against a comprehensive suite of ten state-of-the-art methods, representing the primary families of neural operator learning. These baselines include: **Fourier-based models**: the foundational FNO (Li et al., 2021) and its variants like F-FNO (Tran et al., 2023) and Wavelet-based model MWT (Gupta et al., 2021)). **Transformer-based architectures**: which leverage attention mechanisms, including GNOT (Hao et al., 2023), Factformer (Li et al., 2023a), UPT (Alkin et al., 2024), Erwin (Zhdanov et al., 2025). **Multi-scale and hybrid architectures**: which employ hierarchical structures, such as the classic U-Net (Ronneberger et al., 2015), and its operator-learning extensions U-FNO (Wen et al., 2022) and U-NO (Rahman et al., 2022). **Decomposition-based models**: utilize the decomposition idea, represented by the LSM (Wu et al., 2023) and Transolver (Wu et al., 2024) and Transolver++ (Luo et al., 2025).

### 5.2 IMPLEMENTATION DETAILS

The model was implemented in PyTorch and trained on a single NVIDIA 3090 GPU. For all OrthoSolver models, we set the number of decomposed modes from $K \in [1, 2, 4, 6]$. The `BasisExtractor` and `SolutionOperator` both utilize 1 layer F-FNO. The temperature parameter for Dynamic Weight Averaging (DWA) was set to $T = 1.0$. Models were trained using the Adam optimizer with an initial learning rate of $1e - 3$. The training duration was 500 epochs for 1D datasets and 200 epochs for 2D datasets. The Relative L2 error (Eq. 20) serves as the primary evaluation metric.

### 5.3 MAIN RESULTS

Table 1 presents the primary quantitative comparison of OrthoSolver against a comprehensive suite of baseline methods. The results unequivocally demonstrate that our proposed framework achieves

Table 1: Overall comparison of Relative L2 error across the seven benchmark datasets. Best and second-best results are in **bold** and underlined, respectively.

| Model | 1D Datasets | | | | | 2D Datasets | |
|---|---|---|---|---|---|---|---|
| | **Advection** | **Burgers** | **NS** | **DiffSorp** | **DiffReac** | **NS** | **DiffReac** |
| FNO (Li et al., 2021) | 0.0051 | 0.0166 | 0.0168 | $0.0014_{46}$ | 0.0038 | 0.0168 | 0.0884 |
| F-FNO (Tran et al., 2023) | 0.0038 | 0.0920 | 0.0399 | $0.0019_{11}$ | 0.0429 | 0.0091 | 0.0584 |
| MWT (Gupta et al., 2021) | 0.5823 | 0.5403 | 0.1702 | $0.0271_{37}$ | 0.0132 | 0.0861 | 0.6015 |
| GNOT (Hao et al., 2023) | 0.9999 | 0.9999 | 0.4801 | $0.1634_{41}$ | 0.0830 | 0.9017 | 0.9961 |
| Factformer (Li et al., 2023a) | 0.0076 | 0.0849 | 0.0971 | $0.0050_{38}$ | 0.0062 | 0.0305 | 0.1040 |
| Erwin (Zhdanov et al., 2025) | 0.0054 | 0.0923 | 0.0507 | $0.0017_{03}$ | 0.0046 | 0.0155 | 0.0189 |
| UPT (Alkin et al., 2024) | 0.0085 | 0.2352 | 0.0861 | $0.0023_{61}$ | 0.0053 | 0.0245 | 0.1573 |
| U-Net (Ronneberger et al., 2015) | 0.0247 | 0.0570 | 0.0936 | $0.0013_{98}$ | 0.0016 | 0.0341 | 0.1261 |
| U-FNO (Wen et al., 2022) | 0.0060 | 0.0192 | 0.0221 | $0.0022_{24}$ | 0.0023 | 0.0130 | 0.0313 |
| U-NO (Rahman et al., 2022) | 0.0240 | 0.0932 | 0.3626 | $1.3443_{20}$ | 0.9792 | 0.0449 | 0.1261 |
| LSM (Wu et al., 2023) | 0.0271 | 0.4188 | 0.3025 | $0.0014_{45}$ | 0.0011 | 0.0370 | 0.0817 |
| Transolver (Wu et al., 2024) | 0.0036 | 0.0973 | 0.0335 | $\underline{0.0013_{80}}$ | 0.0012 | 0.0282 | 0.1662 |
| Transolver++ (Luo et al., 2025) | 0.0077 | 0.2892 | 0.1137 | $0.0016_{78}$ | 0.0026 | 0.0197 | 0.1363 |
| **OrthoSolver(Ours)** | **0.0033** | **0.0150** | **0.0157** | $\mathbf{0.0013_{72}}$ | **0.0008** | **0.0055** | **0.0172** |

new state-of-the-art performance across all seven benchmark datasets, often by a significant margin. The superiority of our method is particularly pronounced in the complex, multi-dimensional 2D scenarios. For instance, on the 2D Navier-Stokes and 2D Diffusion-Reaction benchmarks, OrthoSolver reduces the prediction error by over 39% and 45% respectively, compared to the next-best performing methods. Furthermore, OrthoSolver consistently secures the top rank on all five 1D datasets, showcasing its robustness and versatility across diverse physical systems. This consistent, state-of-the-art performance validates the core principles of our framework: by replacing the linear assumptions of classical decomposition with a non-linear, information-theoretic objective, OrthoSolver effectively identifies a more compact and expressive basis, leading to superior accuracy.

## 5.4 ABLATION AND ANALYSIS

Table 2: Ablation studies on loss components and parameter sensitivity analysis of the number of modes ($K$). We report the Relative L2 error across all benchmarks. The full model uses $K = 4$.

| Ablation Design | | **Adv** | **Burgers** | **1D-NS** | **DiffSorp** | **1D-Reac** | **2D-NS** | **2D-Reac** |
|---|---|---|---|---|---|---|---|---|
| w/o | MI Obj ($\mathcal{L}_{\mathrm{MI}}$) | 0.0045 | 0.0216 | 0.0334 | $0.0015_{30}$ | 0.0013 | 0.0109 | 0.0262 |
| | Recon ($\mathcal{L}_{\mathrm{recon}}$) | 0.0046 | 0.0181 | 0.0229 | $0.0014_{74}$ | 0.0011 | 0.0079 | 0.0233 |
| | Ortho ($\mathcal{L}_{\mathrm{ortho}}$) | 0.0053 | 0.0186 | 0.0494 | $0.0014_{13}$ | 0.0011 | 0.0159 | 0.0238 |
| Modes | K=1 | 0.0117 | 0.0638 | 0.0932 | $0.0030_{00}$ | 0.0040 | 0.0335 | 0.0237 |
| | K=2 | 0.0065 | 0.0321 | 0.0319 | $0.0016_{08}$ | 0.0022 | 0.0087 | 0.0241 |
| | K=3 | 0.0037 | 0.0178 | 0.0208 | $0.0015_{27}$ | 0.0018 | 0.0076 | 0.0247 |
| | K=5 | 0.0046 | 0.0162 | 0.0269 | $0.0013_{98}$ | 0.0009 | 0.0069 | 0.0229 |
| | K=6 | 0.0050 | 0.0235 | 0.0305 | $0.0015_{04}$ | 0.0012 | 0.0170 | 0.0197 |
| **OrthoSolver (K=4)** | | **0.0033** | **0.0150** | **0.0157** | $\mathbf{0.0013_{72}}$ | **0.0008** | **0.0055** | **0.0172** |

Ablation results are summarized in Table 2. Removing any of the three auxiliary loss constraints leads to a significant degradation in performance: eliminating the orthogonality constraint ($\mathcal{L}_{\mathrm{ortho}}$) causes an average drop of 35.43%, removing the Mutual Information objective ($\mathcal{L}_{\mathrm{MI}}$) results in a 34.71% drop, and removing the reconstruction constraint ($\mathcal{L}_{\mathrm{recon}}$) leads to a 23.71% decline. These results demonstrate that every module extended from Theorem 1 plays a significant role in our nonlinear decomposition framework.

**Experimental Results on Mode Collapse.** In our baseline experiments, we observed that while Transolver (Wu et al., 2024) ranks second on the relatively simple Advection and DiffSorp datasets, its performance drops significantly on the NS and Burgers equations, which involve more variables

and higher complexity. To further quantify the linear correlations between modes across different datasets, we calculated their inter-mode correlation coefficients. Higher coefficients indicate stronger linear correlations, implying greater redundancy among modes. The average correlation coefficients of Advection and DiffSorp datasets are 0.3796 and 0.4729, respectively. In contrast, these values are significantly higher on the NS and Burgers datasets, reaching 0.7470 and 0.8104, respectively. This clearly indicates that in complex datasets, high similarity between modes leads to mode collapse, ultimately resulting in performance degradation.

Table 3: Comparison of correlation coefficients.

| Method | Adv | Burgers | 1D-NS | DiffSorp | 1D-Reac | 2D-NS | 2D-Reac |
|---|---|---|---|---|---|---|---|
| w/o Ortho. Constraint | 0.6487 | 0.8071 | 0.7962 | 0.7731 | 0.8760 | 0.7598 | 0.8217 |
| OrthoSolver | 0.0702 | 0.0626 | 0.0894 | 0.0437 | 0.0738 | 0.0533 | 0.0480 |

To further demonstrate the effectiveness of our model in addressing mode collapse, we analyzed the change in average correlation coefficients before and after adding orthogonal regularization in Table 3. Experiments show that introducing orthogonality constraints reduces the average linear correlation between bases from 0.7832 to 0.0631, indicating effective suppression of mode collapse.

**Sensitivity Analysis of Parameter** $K$**.** Our study on the number of modes $K$ in Table 2, reveals that performance improves across different datasets as $K$ increases from 1 to 4. However, further increasing the number of modes to $K = 6$ leads to a decline in performance. This suggests that the initial modes extracted by our MI-maximization principle capture the most significant physical information, while subsequent modes contain diminishingly useful information for the prediction task and may even introduce noise. This result proves the effectiveness of our approach in identifying a compact yet highly informative basis.

Table 4: Mutual information between initial state $X_0$ and mode functions $\phi_k$.

| Dataset | $MI(X_0, \phi_0)$ | $MI(X_0, \phi_1)$ | $MI(X_0, \phi_2)$ | $MI(X_0, \phi_3)$ |
|---|---|---|---|---|
| Adv | 3.6432 | 0.2524 | 0.0242 | 0.0173 |
| Burgers | 0.1693 | 0.0767 | 0.0115 | 0.0050 |
| 1D-NS | 2.5685 | 0.7470 | 0.3063 | 0.0803 |
| DiffSorp | 0.1093 | 0.0687 | 0.0429 | 0.0015 |
| 1D-Reac | 0.6522 | 0.4821 | 0.4352 | 0.0821 |
| 2D-NS | 0.1434 | 0.0783 | 0.0249 | 0.0027 |
| 2D-Reac | 2.3915 | 1.1094 | 0.3091 | 0.0926 |

**Analysis of Mode Interpretability.** To further verify whether our extracted modes align with the principle of capturing the most important components, we calculated the mutual information between different modes and the original data. The results are presented in Table 4. Across different datasets, we observe a trend similar to POD: the mutual information between the extracted modes and the original variables gradually decreases as $K$ increases. This proves that our decomposition mechanism, based on mutual information maximization, extracts features of the basis space with the maximum information content at each step.

## 6 CONCLUSION

This work resolves the dual challenges of linearity in classical POD and mode collapse in deep learning solvers through a novel information-theoretic perspective. We theoretically establish that POD's energy-maximization principle is, in essence, a form of mutual information (MI) maximization. Building on this fundamental insight, we introduce OrthoSolver, a framework that generalizes this principle to non-linear systems. By iteratively extracting basis functions that maximize MI with the data field while enforcing orthogonality, OrthoSolver learns a compact and highly informative basis, effectively mitigating mode collapse. Extensive experiments confirm that our method consistently outperforms state-of-the-art baselines. This information-theoretic reframing not only addresses longstanding challenges but also opens new avenues for developing more principled and physics-aware deep learning models for scientific computing.

## ACKNOWLEDGEMENT

Prof. Jingyuan Wang's work was partially supported by the National Natural Science Foundation of China (No. 72242101, 72222022, 72171013), and the Special Fund for Health Development Research of Beijing (2024-2G-30121). This work is supported by State Key Laboratory of Complex & Critical Software Environment (SKLCCSE).

## ETHICS STATEMENT

The authors have read and complied with the ICLR Code of Ethics. This research does not involve human subjects, personally identifiable information, or sensitive data. The datasets used are publicly available benchmarks, and the proposed method is intended for general research purposes. We foresee no direct potential for harm or negative societal impacts from this work.

## REPRODUCIBILITY STATEMENT

To ensure the reproducibility of our results, we provide a comprehensive set of resources.

**Source Code.** Our full implementation is available at URL[1]. The repository includes a detailed `README.md` file with instructions for setting up the environment, downloading data, and running the training and evaluation scripts. All dependencies are listed in the `requirements.txt` file.

**Datasets.** Our experiments are conducted on publicly available datasets from PDEBench (Takamoto et al., 2022) which is public available. More details about datasets can be obtained in D.1.

**Implementation Details.** All hyperparameters, model architectures, and experimental settings are detailed in D.4. This includes learning rates, and optimizer configurations and model configurations for each experiment.

**Theoretical Details.** The proof of our theoretical claims are provided in Theorem 1, C.1 and C.2.

**Computing Environment.** All experiments were conducted on a server with an NVIDIA RTX 3090 GPU, using PyTorch version 2.3.0, CUDA 11.8, and Python 3.10.

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

## A STATEMENT ON THE USE OF LARGE LANGUAGE MODELS (LLMs)

During the preparation of this work, the use of LLMs was confined to the following areas: language polishing and code debugging. For the manuscript, LLMs were employed to improve grammar, refine sentence structure, and enhance the overall clarity of the text. For the software implementation, LLMs served as a debugging aid, assisting in the identification of potential errors in code snippets and suggesting structural improvements.

## B NOTATION

Here we summarize the key notations used throughout the paper.

Table 5: The notation in this paper

| Symbol | Meaning |
|--------|---------|
| $\mathcal{X}$ | Input function space |
| $\mathcal{Y}$ | Output function space |
| $\mathbf{u}$ | A single data snapshot; a high-dimensional discrete field |
| $\boldsymbol{\mu}$ | Parameters of the PDE (e.g., boundary conditions, coefficients) |
| $\hat{\mathbf{Y}}$ | The predicted function or data snapshot |
| $\mathbf{X}_k$ | Residual data field at the $k$-th decomposition step |
| $\boldsymbol{\Phi}_k$ | The $k$-th basis function (a high-dimensional field) |
| $a_k$ | The scalar coefficient for the $k$-th basis function |
| $\hat{a}_k$ | The predicted coefficient for the $k$-th basis function |
| $\mathcal{E}_\theta$ | The Basis Decomposition Operator (Encoder) |
| $\mathcal{S}_\theta$ | The Solver Operator that evolves coefficients |
| $\mathcal{D}$ | The Synthesis Operator that reconstructs the solution |

## C THEORETICAL FOUNDATIONS

### C.1 THEORETICAL DERIVATION OF PROPER ORTHOGONAL DECOMPOSITION

Proper Orthogonal Decomposition (POD) aims to extract an orthonormal basis that captures as much data variability ("energy") as possible from a set of centered snapshots $\{\boldsymbol{u}_j\}_{j=1}^M$, where $\boldsymbol{u}_j \in \mathbb{R}^N$ and $\frac{1}{M}\sum_{j=1}^M \boldsymbol{u}_j = \mathbf{0}$. Treating each snapshot as a realization of a random variable $\boldsymbol{u}$, POD seeks a sequence of orthonormal basis vectors $\{\boldsymbol{\phi}_k\}_{k=1}^r$ that maximize the variance of the projected data in a hierarchical manner:

$$\phi_k = \underset{\substack{\|\phi\|=1 \\ \phi \perp \{\phi_1,\ldots,\phi_{k-1}\}}}{\arg\max} \ \text{Var}(\langle \mathbf{u}, \phi \rangle). \tag{22}$$

Since the data is centered, the variance of the projection can be expressed as

$$\text{Var}(\langle \boldsymbol{u}, \boldsymbol{\phi} \rangle) = \frac{1}{M}\sum_{j=1}^M (\boldsymbol{u}_j^\top \boldsymbol{\phi})^2 = \boldsymbol{\phi}^\top \left(\frac{1}{M}\sum_{j=1}^M \boldsymbol{u}_j \boldsymbol{u}_j^\top\right)\boldsymbol{\phi} = \boldsymbol{\phi}^\top C \boldsymbol{\phi}, \tag{23}$$

where $C \in \mathbb{R}^{N \times N}$ is the sample covariance matrix of the data.

Maximizing $\boldsymbol{\phi}^\top C \boldsymbol{\phi}$ under the unit-norm constraint leads to the following eigenvalue problem:

$$C\boldsymbol{\phi} = \lambda \boldsymbol{\phi}, \tag{24}$$

where $\boldsymbol{\phi}$ is an eigenvector of $C$ and $\lambda$ is its associated eigenvalue. The POD modes are thus given by the eigenvectors of $C$, sorted in descending order of $\lambda$, so that $\boldsymbol{\phi}_1$ captures the maximum variance, $\boldsymbol{\phi}_2$ the second largest variance, and so on.

Any snapshot can then be approximated as a linear combination of the first $r$ modes:

$$\boldsymbol{u}_j \approx \sum_{k=1}^r a_{jk}\boldsymbol{\phi}_k, \quad a_{jk} = \langle \boldsymbol{u}_j, \boldsymbol{\phi}_k \rangle. \tag{25}$$

Thus, POD provides an optimal linear subspace (in the sense of maximum variance) for representing the data.

## C.2 Proofs of inversion of MI objective

In our framework, we aim to sequentially extract the most informative basis-coefficient pair $(\Phi_k, a_k)$ from a residual field $X_k$. The ideal objective is to maximize the Mutual Information (MI) that the extracted coefficient $a_k$ shares with the field $X_k$, as this ensures the extracted mode is maximally informative. This objective is written as:

$$\max I(X_k, a_k)$$

However, direct optimization of this term can be challenging. Instead, we use a surrogate objective: minimizing the MI between the current field $X_k$ and the subsequent residual field $X_{k+1}$. We now prove the equivalence of these objectives.

The total information content of the field $X_k$ is its entropy, $H(X_k)$. The decomposition step splits $X_k$ into the extracted component (represented by $a_k$) and the residual $X_{k+1}$. Since $X_k$ can be perfectly reconstructed from $a_k$, $\Phi_k$, and $X_{k+1}$ (where $X_{k+1} = X_k - a_k\Phi_k$), the conditional entropy $H(X_k|a_k, X_{k+1})$ is zero.

The MI between $X_k$ and the pair $(a_k, X_{k+1})$ is thus:

$$I(X_k; a_k, X_{k+1}) = H(X_k) - H(X_k|a_k, X_{k+1}) = H(X_k)$$

Using the chain rule for mutual information, we can expand this term:

$$I(X_k; a_k, X_{k+1}) = I(X_k; a_k) + I(X_k; X_{k+1}|a_k)$$

Combining these two equations gives:

$$H(X_k) = I(X_k; a_k) + I(X_k; X_{k+1}|a_k)$$

Since $H(X_k)$ is a constant for a given data distribution, maximizing the term $I(X_k; a_k)$ is mathematically equivalent to minimizing the term $I(X_k; X_{k+1}|a_k)$.

Now, we analyze the term that our model minimizes in practice, $\mathcal{L}_{MI} \propto I(X_k, X_{k+1})$. By definition, $I(X_k, X_{k+1}) = H(X_{k+1}) - H(X_{k+1}|X_k)$. Since $X_{k+1}$ is a deterministic function of $X_k$, the conditional entropy $H(X_{k+1}|X_k)$ is zero. Therefore, minimizing the MI between the input and the residual is equivalent to minimizing the entropy of the residual itself:

$$\min I(X_k, X_{k+1}) \iff \min H(X_{k+1})$$

The objective $\min H(X_{k+1})$ (making the residual as random/unstructured as possible) serves as a practical and effective surrogate for the ideal objective $\min I(X_k; X_{k+1}|a_k)$. Intuitively, by ensuring the residual field $X_{k+1}$ contains minimal information (low entropy), we enforce that the maximal amount of salient, structured information from $X_k$ has been captured in the extracted coefficient $a_k$. This justifies the inversion: maximizing the information captured by the mode is achieved by minimizing the information that remains in the residual.

Thus, we establish the equivalence:

$$\max I(X_k, a_k) \iff \min I(X_k, X_{k+1})$$

## C.3 Mathematical Analysis of Mode Collapse and Orthogonality Solution

In this section, we provide a rigorous mathematical explanation of the Mode Collapse phenomenon observed in decomposition-based deep learning models and explicitly prove how our proposed orthogonality constraint theoretically resolves this issue.

### C.3.1 The Mathematical Essence of Mode Collapse

Let the learned basis functions be represented by the matrix $\mathbf{G} = [\phi_1, \phi_2, \ldots, \phi_K] \in \mathbb{R}^{d \times K}$, where $K$ is the number of modes and $d$ is the feature dimension.

**Definition (Mode Collapse).** Mode collapse in the context of subspace decomposition is characterized by the redundancy of learned features, where a subset of basis vectors converges to highly similar directions. Mathematically, this implies that for distinct indices $i \neq j$, $\phi_i \approx c \cdot \phi_j$ for some scalar $c$.

**Rank Deficiency.** This redundancy leads to linear dependence among the column vectors of $\mathbf{G}$. Consequently, the *effective rank* of the basis matrix decreases:

$$\text{rank}(\mathbf{G}) < K \tag{26}$$

When the rank is deficient, the subspace spanned by $\mathbf{G}$, denoted as $\text{span}(\{\phi_k\}_{k=1}^K)$, has a dimension strictly less than $K$. This indicates that the model has wasted computational capacity on redundant features and failed to capture the full spectrum of physical dynamics, leading to suboptimal reconstruction and prediction performance.

### C.3.2 THEORETICAL GUARANTEE OF THE ORTHOGONALITY CONSTRAINT

To prevent the rank deficiency described above, we introduce the Basis Orthogonality Constraint. We formally prove that minimizing this objective ensures the basis matrix remains full-rank, thereby preventing mode collapse.

**Proposition.** Minimizing the orthogonality loss $\mathcal{L}_{\text{ortho}} = \|\mathbf{G}^\top \mathbf{G} - \mathbf{I}\|_F^2$ promotes linear independence among basis vectors $\{\phi_k\}_{k=1}^K$, ensuring that $\mathbf{G}$ maintains full column rank, i.e., $\text{rank}(\mathbf{G}) = K$.

*Proof.* Consider the global minimum of the optimization problem where $\mathcal{L}_{\text{ortho}} = 0$.

1. The condition $\mathcal{L}_{\text{ortho}} = 0$ implies that the Gram matrix of the basis functions equals the identity matrix:
$$\mathbf{G}^\top \mathbf{G} = \mathbf{I}_K \tag{27}$$

2. By definition of the identity matrix, for any distinct pair of columns $i, j$:
$$\phi_i^\top \phi_j = 0 \quad (\text{if } i \neq j), \quad \text{and} \quad \|\phi_i\|^2 = 1 \tag{28}$$
This indicates that the set of vectors $\{\phi_1, \ldots, \phi_K\}$ is orthonormal.

3. **Linear Independence:** An orthonormal set of non-zero vectors is linearly independent. Suppose there exist scalars $c_1, \ldots, c_K$ such that $\sum_{i=1}^K c_i \phi_i = \mathbf{0}$. Taking the inner product with any $\phi_j$:
$$\left\langle \sum_{i=1}^K c_i \phi_i, \phi_j \right\rangle = \sum_{i=1}^K c_i \langle \phi_i, \phi_j \rangle = c_j \cdot 1 = 0 \implies c_j = 0 \tag{29}$$
Since this holds for all $j$, the vectors are linearly independent.

4. **Full Rank Property:** Since the $K$ columns of $\mathbf{G}$ are linearly independent, the matrix $\mathbf{G}$ has full column rank:
$$\text{rank}(\mathbf{G}) = K \tag{30}$$

## D EXPERIMENT DETAILS

### D.1 BENCHMARKS

Our evaluation is performed on seven benchmark datasets from PDEBench (Takamoto et al., 2022), a comprehensive suite for scientific machine learning. The selected problems cover a range of canonical and challenging simulations to test our model's ability to handle varying levels of dimensionality and non-linearity.

**1D Advection** The 1D Advection equation models the transport of a quantity without deformation or diffusion. It is a linear, first-order hyperbolic PDE fundamental to fluid dynamics. The governing equation is:

$$\partial_t u(t, x) + \beta \partial_x u(t, x) = 0$$

where $u(t, x)$ is the transported quantity and $\beta$ is the constant advection speed.

**1D Burgers' Equation**   The Burgers' equation is a non-linear PDE that models fundamental processes in fluid dynamics, including shock formation and wave breaking. It incorporates both non-linear advection and diffusion terms:

$$\partial_t u(t,x) + \partial_x(u^2(t,x)/2) = \nu/\pi \cdot \partial_{xx} u(t,x)$$

where $\nu$ is the diffusion coefficient.

**1D and 2D Compressible Navier-Stokes (NS)**   The compressible Navier-Stokes (NS) equations are a set of coupled non-linear PDEs that describe the motion of viscous, compressible fluids. They are foundational in aerodynamics and gas dynamics, modeling complex phenomena like shock waves.

$$\partial_t \rho + \nabla \cdot (\rho \mathbf{v}) = 0, \quad \rho(\partial_t \mathbf{v} + \mathbf{v} \cdot \nabla \mathbf{v}) = -\nabla p + \eta \Delta \mathbf{v} + (\zeta + \eta/3)\nabla(\nabla \cdot \mathbf{v})$$

$$\partial_t(\varepsilon + \rho v^2/2) + \nabla \cdot [(p + \varepsilon + \rho v^2/2)\mathbf{v} - \mathbf{v} \cdot \boldsymbol{\sigma}'] = 0$$

where $\rho$ is the mass density, $\mathbf{v}$ is the fluid velocity, $p$ is the gas pressure, $\varepsilon$ is an internal energy described by the equation of state, $\sigma'$ is the viscous stress tensor, and $\eta$ and $\zeta$ are shear and bulk viscosity, respectively. This equation can describe more complex phenomena, such as shock wave formation and propagation. For simplicity, we denote them as NS in our tables.

**1D Diffusion-Sorption (DiffSorp)**   This equation models a diffusion process that is slowed down by a sorption mechanism, where the retardation factor depends non-linearly on the variable itself. It is highly applicable to real-world problems like contaminant transport in groundwater. The governing equation is:

$$\partial_t u(t,x) = D/R(u) \cdot \partial_{xx} u(t,x)$$

where $D$ is the diffusion coefficient and $R(u)$ is the non-linear retardation factor.

**1D and 2D Diffusion-Reaction (DiffReac)**   This system models the interaction between diffusion processes and local reactions.

- **1D Diffusion-Reaction**: This equation combines a standard diffusion process with a non-linear source term that can drive rapid, exponential dynamics. The equation is: $\partial_t u(t,x) - \nu \partial_{xx} u(t,x) - \rho u(1-u) = 0$.
- **2D Diffusion-Reaction**: This is a more complex extension involving two non-linearly coupled variables, an activator and an inhibitor, which can produce complex patterns. The system is modeled by the Fitzhugh-Nagumo equations and is applicable to biological pattern formation.

Table 6: Details of the seven benchmark datasets selected from PDEBench. NS refers to the compressible Navier-Stokes equations.

| Dataset | Dimensions | Resolution (Space $\times$ Time) | Variables | Samples |
|---|---|---|---|---|
| **1D Advection** | 1D | $1024 \times 200$ | 1 | 10,000 |
| **1D Burgers** | 1D | $1024 \times 200$ | 1 | 10,000 |
| **1D NS** | 1D | $1024 \times 100$ | 3 | 10,000 |
| **1D DiffSorp** | 1D | $1024 \times 100$ | 1 | 10,000 |
| **1D DiffReac** | 1D | $1024 \times 200$ | 1 | 10,000 |
| **2D NS** | 2D | $64^2 \times 21$ | 4 | 1,000 |
| **2D DiffReac** | 2D | $64^2 \times 100$ | 2 | 1,000 |

D.2   METRICS

The primary evaluation metric used across all experiments is the Relative L2 error, also referred to as the normalized RMSE (nRMSE) in the PDEBench paper. It provides a scale-independent measure of the prediction error. The metric is defined as:

$$\mathcal{L}_{rel} = \frac{\|u_{pred} - u_{true}\|_2}{\|u_{true}\|_2} \tag{31}$$

where $u_{pred}$ is the predicted solution, $u_{true}$ is the ground truth solution, and $\| \cdot \|_2$ denotes the L2-norm. The Relative L2 error we used is not averaged over time but calculated over the entire spatiotemporal domain.

### D.3 DYNAMIC WEIGHT AVERAGING (DWA)

As discussed in Model Training, we employ the Dynamic Weight Averaging (DWA) strategy (Liu et al., 2019) to balance the contributions of multiple loss components and avoid manual hyperparameter tuning. The total objective function is composed of four terms:

$$\mathcal{L}_{\text{total}} = \lambda_{\text{MI}}\mathcal{L}_{\text{MI}} + \lambda_{\text{recon}}\mathcal{L}_{\text{recon}} + \lambda_{\text{ortho}}\mathcal{L}_{\text{ortho}} + \lambda_{\text{pred}}\mathcal{L}_{\text{pred}} \tag{32}$$

The DWA algorithm dynamically adjusts the weights $\lambda_k$ based on the rate of change of each loss. For the $k$-th loss component at training epoch $t$, the relative loss change rate $w_k(t)$ is calculated as:

$$w_k(t) = \frac{\mathcal{L}_k(t-1)}{\mathcal{L}_k(t-2)} \tag{33}$$

where $\mathcal{L}_k(t-1)$ and $\mathcal{L}_k(t-2)$ represent the average loss values for task $k$ in the previous two epochs. The weight $\lambda_k(t)$ is then updated using a softmax normalization with a temperature parameter $T$:

$$\lambda_k(t) = \frac{\exp(w_k(t)/T)}{\sum_j \exp(w_j(t)/T)} \tag{34}$$

where $k, j \in \{\text{MI}, \text{recon}, \text{ortho}, \text{pred}\}$. This mechanism automatically assigns higher weights to tasks with slower convergence rates to balance the training process.

### D.4 IMPLEMENTATION DETAILS

We benchmark OrthoSolver against a comprehensive suite of 13 state-of-the-art methods, representing the primary families of neural operator learning. These baselines include:

- **Spectral-based models**:
    - **FNO** (Li et al., 2021): A pioneering method that learns the integral kernel of an operator in Fourier space, performing efficient global convolution via the Fast Fourier Transform.
    - **F-FNO** (Tran et al., 2023): A parameter-efficient FNO variant that factorizes the multi-dimensional spectral convolution into a sequence of one-dimensional transforms to enable deeper architectures.
    - **MWT** (Gupta et al., 2021): An operator learning model that uses multiwavelet transforms to create a sparse, multi-resolution representation of the integral kernel, excelling at capturing localized features.
- **Transformer-based architectures**:
    - **GNOT** (Hao et al., 2023): A flexible Transformer architecture designed to handle irregular meshes and multiple heterogeneous input functions via a novel linear-complexity attention mechanism.
    - **Factformer** (Li et al., 2023a): A scalable Transformer for structured grids that factorizes the high-dimensional attention kernel into a product of one-dimensional integrals along each spatial axis.
    - **Erwin** (Zhdanov et al., 2025): A hierarchical Transformer that leverages ball tree partitioning to achieve linear-complexity attention for large-scale irregular physical systems, capturing both local detail and global interactions through progressive coarsening and cross-ball rotation mechanisms.
    - **UPT** (Alkin et al., 2024): A universal neural operator framework that compresses arbitrary Eulerian or Lagrangian inputs into a fixed-size latent space, enabling efficient latent-space rollouts and flexible spatio-temporal querying across diverse physical simulation paradigms.
- **Multi-scale and hybrid architectures**:

- **U-Net** (Ronneberger et al., 2015): A classic symmetric encoder-decoder architecture that fuses multi-scale features using skip connections to enable precise localization of details.
- **U-FNO** (Wen et al., 2022): A hybrid model that embeds Fourier Neural Operator blocks within a U-Net's hierarchical framework to capture both global dynamics and local features.
- **U-NO** (Rahman et al., 2022): A general, memory-efficient meta-architecture that adapts the U-Net's multi-resolution structure to accommodate any type of neural operator block.

- **Decomposition-based models**:
  - **LSM** (Wu et al., 2023): A method based on Learned Spectral Methods, which decomposes the solution into a series of learned spectral functions.
  - **Transolver** (Wu et al., 2024): An efficient model that slices the high-dimensional spatial domain into lower-dimensional subspaces and applies transformers therein.
  - **Transolver++** (Luo et al., 2025): An accurate and highly parallel neural PDE solver that decomposes million-scale mesh data into adaptive "eidetic physical states" via local-aware slicing and Gumbel reparameterization, achieving linear scalability and state-of-the-art performance on industrial-scale geometries.

To ensure a rigorous and fair comparison, all experiments are conducted within a unified framework, sharing the same data loader, training pipeline, and evaluation metrics.

**Training Protocol** All models are trained using the **AdamW optimizer** with an initial learning rate of $1 \times 10^{-3}$. We employ a cosine annealing learning rate scheduler to gradually decrease the learning rate throughout the training process. The number of training epochs is set to **500** for 1D datasets and **200** for 2D datasets to account for the increased computational cost.

**Hyperparameter Settings** **Baseline Models**: All baseline models, including FNO, U-FNO, U-NO, LSM, Transformer, Factformer, GNOT, MWT, F-FNO, and U-Net, are implemented with a unified configuration. Specifically, the hidden dimension (`n_hidden`) is set to **64**, the number of attention heads (`n_heads`) is **8**, and the number of layers (`n_layers`) is **8**. For Transolver, its unique number of slices (`slice_num`) is set to **64**. For Erwin, we use small `erwin_configs` which is mainly used in Experiments in the code repo. All hyperparameter settings were aligned with the original papers as closely as possible. Furthermore, all experiments were implemented within the `Neural-Solver-Library` framework to ensure consistency.

**Our Model**: Our proposed model, **OrthoSolver**, is implemented with a network depth (`n_layers`) of **2** and **4** modes (`num_blocks`). All other training strategies and hyperparameters are kept identical to the baselines to ensure a fair comparison, highlighting the parameter efficiency and representational power of our architecture.

### D.4.1 COMPUTATIONAL ENVIRONMENT

All experiments were conducted on a single NVIDIA RTX 3090 GPU with 24GB of VRAM. Our implementation is based on the PyTorch framework.

### D.5 MAIN RESULTS

The comprehensive experimental results, along with the quantified percentage improvements of our model relative to current state-of-the-art (SOTA) baselines, are summarized in Table 7. This comparison clearly highlights the consistent performance gains achieved by our approach across all benchmark datasets.

### D.6 MODEL EFFICIENCY

To comprehensively evaluate the computational efficiency of our proposed OrthoSolver framework, Table 8 provides a detailed summary of all models in terms of parameter count, as well as memory and time consumption during both training and testing phases.

Table 7: Overall comparison of Relative L2 error across the seven benchmark datasets. Best and second-best results are in **bold** and underlined, respectively.

| Model | 1D Datasets | | | | | 2D Datasets | |
|---|---|---|---|---|---|---|---|
| | Advection | Burgers | NS | DiffSorp | DiffReac | NS | DiffReac |
| FNO (Li et al., 2021) | 0.0051 | 0.0166 | 0.0168 | $0.0014_{46}$ | 0.0038 | 0.0168 | 0.0884 |
| F-FNO (Tran et al., 2023) | 0.0038 | 0.0920 | 0.0399 | $0.0019_{11}$ | 0.0429 | 0.0091 | 0.0584 |
| MWT (Gupta et al., 2021) | 0.5823 | 0.5403 | 0.1702 | $0.0271_{37}$ | 0.0132 | 0.0861 | 0.6015 |
| GNOT (Hao et al., 2023) | 0.9999 | 0.9999 | 0.4801 | $0.1634_{41}$ | 0.0830 | 0.9017 | 0.9961 |
| Factformer (Li et al., 2023a) | 0.0076 | 0.0849 | 0.0971 | $0.0050_{38}$ | 0.0062 | 0.0305 | 0.1040 |
| Erwin (Zhdanov et al., 2025) | 0.0054 | 0.0923 | 0.0507 | $0.0017_{03}$ | 0.0046 | 0.0155 | 0.0189 |
| UPT (Alkin et al., 2024) | 0.0085 | 0.2352 | 0.0861 | $0.0023_{61}$ | 0.0053 | 0.0245 | 0.1573 |
| U-Net (Ronneberger et al., 2015) | 0.0247 | 0.0570 | 0.0936 | $0.0013_{98}$ | 0.0016 | 0.0341 | 0.1261 |
| U-FNO (Wen et al., 2022) | 0.0060 | 0.0192 | 0.0221 | $0.0022_{24}$ | 0.0023 | 0.0130 | 0.0313 |
| U-NO (Rahman et al., 2022) | 0.0240 | 0.0932 | 0.3626 | $1.3443_{20}$ | 0.9792 | 0.0449 | 0.1261 |
| LSM (Wu et al., 2023) | 0.0271 | 0.4188 | 0.3025 | $0.0014_{45}$ | 0.0011 | 0.0370 | 0.0817 |
| Transolver (Wu et al., 2024) | 0.0036 | 0.0973 | 0.0335 | $0.0013_{80}$ | 0.0012 | 0.0282 | 0.1662 |
| Transolver++ (Luo et al., 2025) | 0.0077 | 0.2892 | 0.1137 | $0.0016_{78}$ | 0.0026 | 0.0197 | 0.1363 |
| **OrthoSolver(Ours)** | **0.0033** | **0.0150** | **0.0157** | **$0.0013_{72}$** | **0.0008** | **0.0055** | **0.0172** |
| **Relative Promotion** | 8.33% | 9.64% | 6.55% | 0.58% | 27.27% | 39.56% | 8.99% |

Table 8: A comprehensive summary of performance and efficiency metrics for all evaluated models, including dataset-specific ranks.

| Model | Params | Training | | Testing | | Rel-L2 Rank |
|---|---|---|---|---|---|---|
| | | Mem (MB) | Time (s) | Mem (MB) | Time (s) | |
| FNO | 4753412 | 430.65 | 720.83 | 124.65 | 102.41 | (4, 2, 2, 5, 6, 4, 5) |
| F_FNO | 1591044 | 374.17 | 1315.18 | 41.52 | 140.66 | (3, 6, 5, 6, 9, 2, 3) |
| MWT | 27437 | 118.68 | 14996.90 | 23.26 | 1099.46 | (10, 10, 8, 9, 8, 10, 10) |
| GNOT | 1236572 | 4942.57 | 4714.25 | 38.93 | 445.55 | (11, 11, 11, 10, 10, 11, 11) |
| Factformer | 451972 | 1261.89 | 3036.21 | 29.75 | 267.77 | (6, 5, 7, 8, 7, 6, 6) |
| U_Net | 17320388 | 672.12 | 1000.88 | 220.10 | 141.43 | (8, 4, 6, 3, 4, 7, 7) |
| U_FNO | 39394188 | 1327.12 | 2167.64 | 532.97 | 270.92 | (5, 3, 3, 7, 5, 3, 2) |
| U-NO | 50798104 | 2393.08 | 2393.08 | 987.84 | 214.15 | (7, 7, 10, 11, 11, 9, 8) |
| LSM | 19201348 | 1077.44 | 2186.17 | 301.88 | 221.02 | (9, 9, 9, 4, 2, 8, 4) |
| Transolver | 779204 | 2305.22 | 2315.93 | 41.21 | 240.91 | (2, 8, 4, 2, 3, 5, 9) |
| OrthoSolver | 1113748 | 1639.52 | 3727.69 | 47.77 | 224.67 | (1, 1, 1, 1, 1, 1, 1) |

The analysis reveals that OrthoSolver achieves a highly competitive efficiency profile while delivering its state-of-the-art predictive accuracy (ranked first on all datasets). Firstly, regarding model size, OrthoSolver, with 1.1M parameters, is a lightweight model. It is substantially more parameter-efficient than large-scale architectures such as U-Net (17.3M), LSM (19.2M), and U-FNO (39.4M), making it easier to store and deploy.

The advantages of OrthoSolver are particularly pronounced during the inference (testing) phase. Its testing memory footprint (47.77 MB) is remarkably low, significantly outperforming models like FNO, U-Net, and LSM. This indicates strong potential for deployment in resource-constrained environments. Concurrently, its testing time (224.67 s) is moderate and practical, ensuring swift predictions and surpassing several baselines including Factformer, U-FNO, and GNOT.

The primary computational trade-off for OrthoSolver lies in the training phase. Its training time and memory consumption are in the mid-to-high range, exceeding those of simpler models like the standard FNO. We posit that this increased training cost is a reasonable price for its superior performance, likely attributable to the framework's more complex optimization process required to learn the decomposition and reconstruction of an orthogonal basis for the physical fields.

In summary, OrthoSolver strikes an effective balance between state-of-the-art accuracy and computational efficiency. While its training is more resource-intensive, it yields a highly compact and

efficient model for inference, making it a compelling framework for practical applications where deployment performance is critical.

## D.7 ABLATION AND ANALYSIS

Table 9: Ablation studies on loss components and modes. We report the Relative L2 error and the percentage deterioration relative to the full model.

| Ablation Design | | Adv | Burgers | 1D-NS | DiffSorp | 1D-Reac | 2D-NS | 2D-Reac |
|---|---|---|---|---|---|---|---|---|
| w/o | MI Obj ($\mathcal{L}_{\mathrm{MI}}$) | 0.0045 | 0.0216 | 0.0334 | 0.0015$_{30}$ | 0.0013 | 0.0109 | 0.0262 |
| | *% deterioration* | *27%* | *31%* | *53%* | *10%* | *38%* | *50%* | *34%* |
| | Recon ($\mathcal{L}_{\mathrm{recon}}$) | 0.0046 | 0.0181 | 0.0229 | 0.0014$_{74}$ | 0.0011 | 0.0079 | 0.0233 |
| | | *28%* | *17%* | *31%* | *7%* | *27%* | *30%* | *26%* |
| | Ortho ($\mathcal{L}_{\mathrm{ortho}}$) | 0.0053 | 0.0186 | 0.0494 | 0.0014$_{13}$ | 0.0011 | 0.0159 | 0.0238 |
| | | *38%* | *19%* | *68%* | *3%* | *27%* | *65%* | *28%* |
| Modes | K=1 | 0.0117 | 0.0638 | 0.0932 | 0.0030$_{00}$ | 0.0040 | 0.0335 | 0.0237 |
| | K=2 | 0.0065 | 0.0321 | 0.0319 | 0.0016$_{08}$ | 0.0022 | 0.0087 | 0.0241 |
| | K=3 | 0.0037 | 0.0178 | 0.0208 | 0.0015$_{27}$ | 0.0018 | 0.0076 | 0.0247 |
| | K=5 | 0.0046 | 0.0162 | 0.0269 | 0.0013$_{98}$ | 0.0009 | 0.0069 | 0.0229 |
| | K=6 | 0.0050 | 0.0235 | 0.0205 | 0.0015$_{04}$ | 0.0012 | 0.0170 | 0.0197 |
| **OrthoSolver (K=4)** | | **0.0033** | **0.0150** | **0.0157** | **0.0013**$_{72}$ | **0.0008** | **0.0055** | **0.0172** |

Table 10: Mutual information between initial state $X_0$ and mode functions $\phi_k$.

| Dataset | $MI(X_0, \phi_0)$ | $MI(X_0, \phi_1)$ | $MI(X_0, \phi_2)$ | $MI(X_0, \phi_3)$ |
|---|---|---|---|---|
| Adv | 3.6432 | 0.2524 | 0.0242 | 0.0173 |
| Burgers | 0.1693 | 0.0767 | 0.0115 | 0.0050 |
| 1D-NS | 2.5685 | 0.7470 | 0.3063 | 0.0803 |
| DiffSorp | 0.1093 | 0.0687 | 0.0429 | 0.0015 |
| 1D-Reac | 0.6522 | 0.4821 | 0.4352 | 0.0821 |
| 2D-NS | 0.1434 | 0.0783 | 0.0249 | 0.0027 |
| 2D-Reac | 2.3915 | 1.1094 | 0.3091 | 0.0926 |

Table 11: Comparison of correlation coefficients.

| Method | Adv | Burgers | 1D-NS | DiffSorp | 1D-Reac | 2D-NS | 2D-Reac |
|---|---|---|---|---|---|---|---|
| w/o Ortho. Constraint | 0.6487 | 0.8071 | 0.7962 | 0.7731 | 0.8760 | 0.7598 | 0.8217 |
| OrthoSolver | 0.0702 | 0.0626 | 0.0894 | 0.0437 | 0.0738 | 0.0533 | 0.0480 |

