# OpenReview forum: "OrthoSolver: A Neural Proper Orthogonal Decomposition Solver For PDEs"
_ICLR.cc/2026/Conference — ICLR 2026 Poster_

### Official Review · Reviewer_e7gJ · 2025-10-28

**Soundness:** 3
**Presentation:** 3
**Contribution:** 3
**Rating:** 6
**Confidence:** 4

**Summary:**

The paper introduces OrthoSolver, a Neural Operator architecture that combines efficient decomposition with robust nonlinear modeling by exploiting an information-theoretic perspective on dimensionality reduction. The method introduces two key innovations to the neural operator community: an information-theoretic nonlinear approach for identifying reduced modes (enhance expressivity), and an orthogonality regularization term that prevents redundant feature extraction (reduce mode collapse). The approach is both elegant and theoretically sound. The authors validate their method on five different 1D and two different 2D PDEs, and further investigate the key design choices of their framework through an in-depth ablation study.

**Strengths:**

The paper addresses the problem of combining efficient mode decomposition (to extract relevant features) with robust nonlinear modeling, while simultaneously preventing mode collapse. This problem is relevant as it clearly captures the dichotomy between traditional Reduced Order Model (ROM) techniques, which have solid mathematical foundations but suffer from linearity limitations, and Neural Operator–based data-driven modeling, which offers high nonlinearity but often lacks theoretical soundness. Among the main strengths of the paper, I highlight the following:

1. An original application of mutual information optimization to guide PDE mode decomposition, which could potentially impact other fields such as ROM and manifold learning.
2. A clear and well-structured review of the Neural Operator literature.
3. The introduction of a Basis Orthogonality Constraint to promote diversity among the learned basis functions.
4. A comprehensive experimental section that includes both 1D and 2D PDEs.

The derivation presented in Appendix C.2 appears sound, and the experimental details are sufficiently complete to allow for reproduction of the results.

**Weaknesses:**

My primary concern is robustness and comparison with state-of-the-art models. Given these clarifications in an author response, I would be willing to increase the score.

1. A more in-depth analysis and explanation of the model’s performance (Table 1) are needed. In particular:
   - Important models [1, 2, 3] are missing and should be included as benchmark baselines. The field is evolving rapidly and it is important to compare against the best models.
   - The evaluation metric for Table 1 is not clearly explained — is the L2 error averaged over time?

2. How were the hyperparameters for both your model and the baselines chosen? In the **Hyperparameter Settings** section, this information is not specified. Are the hyperparameters the same as those used in the PDEBench paper’s baseline results (see Tables 2–9 in [4])? Are the results better than the ones reported by PDEBench (in order to claim SOTA).

3. Can you provide variance results across multiple network initialization seeds for the different problems? This is important to ensure that the results are not cherry-picked for a specific initialization and that the methodology is robust.


***References***

[1] Zhdanov, Maksim, Max Welling, and Jan-Willem van de Meent. "Erwin: A tree-based hierarchical transformer for large-scale physical systems." ICML, 2025.

[2] Alkin, B., F  ̈urst, A., Schmid, S., Gruber, L., Holzleitner, M., and Brandstetter, J. Universal physics transformers: A framework for efficiently scaling neural operators. NeurIPS, 2024.

[3] Luo, H., Wu, H., Zhou, H., Xing, L., Di, Y., Wang, J., and Long, M. Transolver++: An accurate neural solver for pdes on million-scale geometries. CoRR, 2025.

[4] Takamoto, Makoto, et al. "Pdebench: An extensive benchmark for scientific machine learning." Advances in Neural Information Processing Systems 35 (2022): 1596-1611.

**Questions:**

1. Are the modes interpretable? For example, in classical POD, the most energetic modes are those that explain the greatest variance. Do we observe a similar behavior here? It would be helpful to plot the modes in order of energy level to assess their interpretability.
2. In Table 2, why does the model without the reconstruction constraint seem to perform almost as well as the full model? I would have expected this constraint to be very important, as it measures the discrepancy between the original function and its reconstruction from the full set of extracted modes.

---

> ### Author Response · Authors · 2025-11-20
> **Response to Reviewer e7gJ [1/2]: Addressing W1 & W2 & W3**
>
> ---
> # W1: Baseline Expansion and L2 Metric Clarification
>
> ## W1.1 Baseline Expansion
>
> Your suggestions regarding baselines are very helpful!
>
> We preliminary conducted experiments on three representative datasets. The complete experimental results are presented in the table below. We achieved better performance than the baselines across all three datasets. We will further complete the experiments for the remaining four datasets and include them in the final version of the paper.
>
> **Table: Comparison of error metrics with baseline methods.**
>
> | Method | Advection | 1D-NS | 2D-NS |
> | :--- | :--- | :--- | :--- |
> | erwin [1] | 0.0054 | 0.0507 | 0.0155 |
> | Transolver++ [2] | 0.0076 | 0.1137 | 0.0197 |
> | UPT [3] | 0.0085 | 0.0861 | 0.0245 |
> | **OrthoSolver (Ours)** | **0.0033** | **0.0157** | **0.0055** |
>
> ## W1.2 Metric Clarification
>
> Your suggestion to clarify the metric is very meaningful！
>
> The Relative L2 error we used is not averaged over time but calculated over the entire spatiotemporal domain (as defined in Eq. 20 of the paper). This calculation method is consistent with that used in papers [2], [4], and [5].
>
> Additionally, our overall code framework is implemented based on the **Neural-Solver-Library**, ensuring that all models are compared under the same standards.
>
> We will optimize the explanation of this section in the final version of the paper.
>
> ---
> # W2: Hyperparameter Setting And PDEBench Comparison
>
> Thank you for your suggestion to elaborate on hyperparameter settings. Clarifying this part of the experimental setup is important.
>
> ## Hyperparameter Setting
>
> For OrthoSolver, the number of modes $K$ is set to 4, the learning rate $lr=1e-3$, and the optimizer is AdamW. This setting was chosen based on parameter sensitivity experiments, which showed best performance at $K=4$ (please refer to the experimental analysis in Section 5.4).
>
> To maintain consistency with the Baselines, we referred to the original papers and the hyperparameter settings in the Neural-Solver-Library code repository. For example, Transolver's `slice_num` is set to 64, and F-FNO's `n_layers` is set to 8, etc.
>
> Thank you again for your suggestion. Detailed Baseline and hyperparameter settings can be found in Appendix D.3 under **Training Protocol and Hyperparameter Settings** and in the code repository mentioned in the **REPRODUCIBILITY STATEMENT**. Furthermore, we will further supplement and refine this description in the final version of the paper.
>
> ## PDEBench Comparison
>
> For U-Net and FNO implemented in PDEBench [6], we set parameters such as $lr=1e-3$ and $num\\_layers=8$ to remain consistent with PDEBench settings.
>
> Our Relative L2 metric corresponds to the nRMSE in PDEBench (Tables 2-9 in [6]). We compare the results in the table below. The results indicate that OrthoSolver's overall metrics are superior to PDEBench. Here, we selected the best model performance for each dataset from PDEBench for comparison.
>
> **Table: Performance comparison on diverse PDE datasets.**
>
> | Dataset | PDEBench | OrthoSolver |
> | :--- | :--- | :--- |
> | Adev | 0.0093 | **0.0033** |
> | Burgers | 0.0420 | **0.0150** |
> | 1D-CFD | 0.0950 | **0.0157** |
> | ReacDiff | 0.0014 | **0.0008** |
> | Sorp | 0.0078 | **0.0013** |
>
> ---
> # W3: Robustness & Variance
>
> Your suggestion regarding experimental robustness is very important and meaningful. In our previous experiments, we did not actively set a random seed.
>
> Following your advice, we repeated our experiments using five different network initialization seeds to evaluate the mean and variance of the results.
>
> We conducted experiments on three datasets, and the results are as follows:
> * **Advection dataset:** $0.00328 \pm 0.00025$
> * **1D-NS dataset:** $0.01578 \pm 0.00038$
> * **2D-NS dataset:** $0.00548 \pm 0.00025$
>
> Once all experiments are completed, we will add the full variance results to the final paper.
>
> ---
> **References:**
>
> [1] Zhdanov, Maksim, Max Welling, and Jan-Willem van de Meent. "Erwin: A tree-based hierarchical transformer for large-scale physical systems." ICML, 2025.
>
> [2] Alkin, B., Fürst, A., Schmid, S., Gruber, L., Holzleitner, M., and Brandstetter, J. Universal physics transformers: A framework for efficiently scaling neural operators. NeurIPS, 2024.
>
> [3] Luo, H., Wu, H., Zhou, H., Xing, L., Di, Y., Wang, J., and Long, M. Transolver++: An accurate neural solver for pdes on million-scale geometries. ICML, 2025.
> [4] Wu H, Hu T, Luo H, et al. Solving high-dimensional pdes with latent spectral models. ICML, 2023.
>
> [5] Tran A, Mathews A, Xie L, et al. Factorized fourier neural operators. ICLR, 2023.
>
> [6] Takamoto M, Praditia T, Leiteritz R, et al. Pdebench: An extensive benchmark for scientific machine learning. NeurIPS, 2022.

---

> > ### Author Response · Authors · 2025-11-20
> > **Response to Reviewer e7gJ [2/2]: Addressing Q1 & Q2**
> >
> > ---
> > # Q1: Question fo Mode Interpretability
> >
> > Thank you for your constructive feedback on interpretability!
> > This is very helpful for us to deeply understand the decomposition mechanism based on mutual information maximization.
> >
> > We calculated the mutual information between different modes and the original initial state across different datasets. The results are shown in the table below. As you mentioned, similar to POD, it can be observed that the mutual information between the extracted modes and the original variables gradually decreases as $K$ increases.
> >
> > This proves that our decomposition mechanism based on mutual information maximization extracts the features of the basis space with the maximum information content at each step.
> >
> > **Table: Mutual information between initial state $X_0$ and mode functions $\phi_k$ across datasets.**
> >
> > | Dataset | $MI(X_0, \phi_0)$ | $MI(X_0, \phi_1)$ | $MI(X_0, \phi_2)$ | $MI(X_0, \phi_3)$ |
> > | :--- | :--- | :--- | :--- | :--- |
> > | 1D Advection | 3.6432 | 0.2524 | 0.0242 | 0.0173 |
> > | 1D-NS | 0.1093 | 0.0687 | 0.0429 | 0.0015 |
> > | 2D-NS | 0.1434 | 0.0783 | 0.0249 | 0.0027 |
> >
> > Thank you once again sincerely for your constructive feedback regarding the interpretability of the Mode. **We will include additional experimental analysis on this aspect in the final version of the paper** to thoroughly demonstrate the interpretability of our decomposition framework.
> >
> > ---
> > # Q2: Reconstruction Constraint Contribution
> >
> > As you stated, the reconstruction constraint is very important, as it measures the difference between the original function and the mode space reconstruction!
> >
> > We have supplemented the ablation study with the percentage of performance degradation for different settings to assess the impact more clearly. The experimental results show that the model without the reconstruction constraint degrades by approximately **30%** on average compared to the full model. The precise results are shown in the table below, which illustrates the importance of this constraint.
> >
> > **Table: Ablation on reconstruction loss.**
> >
> > | Ablation Design | Advection | 1D-NS | 2D-NS |
> > | :--- | :--- | :--- | :--- |
> > | w/o $\mathcal{L}_{\text{recon}}$ | 0.0046 | 0.0229 | 0.0079 |
> > | OrthoSolver (Full Model=) | **0.0033** | **0.0157** | **0.0055** |
> > | **Relative Drop (%)** | **28.2%** | **31.4%** | **30.3%** |
> >
> > Finally, we will include the percentage of performance degradation for different ablation settings in the final paper to clearly explain the impact of each component.
> >
> > ---
> > **References:**
> >
> > [1] Zhdanov, Maksim, Max Welling, and Jan-Willem van de Meent. "Erwin: A tree-based hierarchical transformer for large-scale physical systems." ICML, 2025.
> >
> > [2] Alkin, B., Fürst, A., Schmid, S., Gruber, L., Holzleitner, M., and Brandstetter, J. Universal physics transformers: A framework for efficiently scaling neural operators. NeurIPS, 2024.
> >
> > [3] Luo, H., Wu, H., Zhou, H., Xing, L., Di, Y., Wang, J., and Long, M. Transolver++: An accurate neural solver for pdes on million-scale geometries. ICML, 2025.
> > [4] Wu H, Hu T, Luo H, et al. Solving high-dimensional pdes with latent spectral models. ICML, 2023.
> >
> > [5] Tran A, Mathews A, Xie L, et al. Factorized fourier neural operators. ICLR, 2023.
> >
> > [6] Takamoto M, Praditia T, Leiteritz R, et al. Pdebench: An extensive benchmark for scientific machine learning. NeurIPS, 2022.

---

> ### Comment · Reviewer_e7gJ · 2025-11-24
> **Response**
>
> I thank the authors for addressing my concerns. I have updated my score, and I'm in favour of the acceptance of the paper.
>
> I find the mode analysis particularly interesting; not many Neural Operator models are interpretable, while OrthoSolver provides a fresh perspective. Also, the paper shows many SOTA results, advancing the field.

---

> ### Author Response · Authors · 2025-11-25
> **Gratitude for Feedback and Score Increase**
>
> Dear Reviewer e7gJ,
>
> We sincerely appreciate your continued engagement and the constructive feedback on our submission.
>
> In particular, we are grateful for your insightful comments regarding modal interpretability and the Reconstruction Constraint, which have helped strengthen our paper. We also deeply appreciate your recognition of our proposed mutual information decomposition framework and your decision to raise the score to 8.
>
> Your insights are highly valuable to us and have significantly contributed to improving our work.
>
> Thank you again for your thoughtful review.

---

### Official Review · Reviewer_wLXC · 2025-11-01

**Soundness:** 3
**Presentation:** 3
**Contribution:** 3
**Rating:** 6
**Confidence:** 4

**Summary:**

The paper builds a connection between POD and mutual information maximization under linear-Gaussian assumptions. Based on this, the authors proposed a data-driven neural proper orthogonal decomposition solver for PDEs. The mutual information is encoded into the total loss function, which will increase the training difficulty. The authors conduct many numerical experiments and many comparisons with other methods to demonstrate the performance of the proposed method.

**Strengths:**

Providing a large amount of the empirical results.

**Weaknesses:**

1. The equivalence between variance and mutual information only holds under the strong linear Gaussian assumption, which is rarely satisfied in nonlinear partial differential equations. This article does not explain why this connection is still meaningful or useful in the highly nonlinear cases it aims to address.
2. Too many penalty terms have to be introduced into the loss function. Is it very hard to optimize?

**Questions:**

Refer to Weaknesses

---

> ### Author Response · Authors · 2025-11-20
> **Response to Reviewer wLXC**
>
> ---
> # W1: Relevance Beyond Linear Assumptions
>
> The question you raised is crucial! In fact, it is precisely our in-depth consideration of this issue that allowed us to generalize the linear framework of POD to more general nonlinear scenarios.
>
> 1. Whether it is classical POD or our OrthoSolver, **the core objective remains the same**: to identify **the dominant basis functions** within the data to represent the physical fields.
>
> 2. Under the assumption of **linear decomposition** of POD, the search for the **"most important"** basis in POD is mathematically derived as **"maximizing variance."** However, variance fundamentally **only measures the second-order moments** of the data. When dealing with highly nonlinear PDEs, relying solely on second-order moments **fails to fully capture complex high-order dependency structures**, thereby limiting the effectiveness of variance metrics in nonlinear contexts [1].
>
> 3. To achieve **decomposition for nonlinear fields**, we approach this from an information-theoretic perspective, using **mutual information** to quantify the importance of basis functions. Mutual information is a more general measure of dependency; it not only encompasses second-order moments but also characterizes higher-order moments and various nonlinear relationships, making it better aligned with the intrinsic structure of nonlinear PDEs.
>
> 4. **Theorem 1** serves as the key to **extending** the POD concept to nonlinear scenarios. This theorem rigorously proves that **under the linear Gaussian assumption, our method naturally degenerates into POD.** This demonstrates that POD is effectively a special case of our method under this specific assumption. Consequently, our approach represents a consistent generalization of POD, further validating its rationality and consistency in more general situations.
>
> 5. Thank you again for your suggestion. In the final version, we will further optimize the relevant descriptions to explicitly emphasize our core contribution of generalizing from linear POD to nonlinear mutual information decomposition.
>
> ---
> # W2: Multi-Objective Optimization
>
> As you mentioned, we introduced multiple sub-task losses into our objective function. The loss function is defined as follows:
>
> $$
> \mathcal{L}\_{total} = \lambda\_{MI} \mathcal{L}\_{MI} + \lambda\_{recon} \mathcal{L}\_{recon} + \lambda\_{ortho} \mathcal{L}\_{ortho} + \lambda\_{pred} \mathcal{L}\_{pred}
> $$
>
>
> Traditional methods require manual tuning of the weights $\lambda$ for different types of losses. This is challenging for optimization, as it requires fine-tuning these weights to avoid training collapse and to balance different losses effectively.
>
> To address this optimization challenge, as mentioned in Section 4.5 (**MODEL TRAINING**), we employed the **Dynamic Weight Averaging (DWA)** [2] multi-task optimization strategy. This approach automatically adjusts weights during the training process based on the rate of change of different task losses. It avoids complex manual parameter tuning and effectively reduces gradient conflicts and training instability caused by multiple tasks/constraints.
>
> The formula for Dynamic Weight Averaging is as follows:
>
> $$\lambda_{i} = \frac{\exp(w_i(t)/T)}{\sum_{i}\exp(w_i(t)/T)}$$
>
> where $w_{i}(t)$ is calculated as:
>
> $$w_{i}(t) = \frac{\mathcal{L}_i(t-1)}{\mathcal{L}_i(t-2)}$$
>
> Here, $i$ corresponds to the different types of losses.
>
> Thank you for your question and suggestion. We will add the DWA formulas to the appendix of the final paper to clearly explain the role of this component.
>
> ---
> **References:**
>
> [1] Demo N, Tezzele M, Rozza G. A DeepONet multi-fidelity approach for residual learning in reduced order modeling[J]. Advanced Modeling and Simulation in Engineering Sciences, 2023.
>
> [2] Liu S, Johns E, Davison A J. End-to-end multi-task learning with attention. CVPR. 2019.
>
> ---

---

> ### Author Response · Authors · 2025-11-25
> **Kind Follow-up and Openness to Further Discussion**
>
> Dear Reviewer wLXC,
>
> We sincerely appreciate your time and the constructive feedback for our submission. We have carefully addressed all of your comments and questions above, and we hope our clarifications adequately resolve the concerns you raised.
>
> If you have the opportunity to review our responses, we would greatly appreciate any further feedback or discussion at your convenience. Your insights are highly valuable to us and have significantly contributed to improving our work.
>
> Thank you again for your thoughtful review.

---

### Official Review · Reviewer_GFif · 2025-11-01

**Soundness:** 3
**Presentation:** 4
**Contribution:** 2
**Rating:** 2
**Confidence:** 3

**Summary:**

The authors introduce a paradigm shift for POD away from optimal energy-based modes to maximizing mutual information among modes. The work also introduces a novel learning architecture for mode decomposition/training in neural PDEs. Strong empirical performance is reported across several PDE benchmarks.

**Strengths:**

- The paper tackles relevant challenges in neural PDEs including principled mode selection and training.
- The work bridges statistical objectives (MI) and reduced-order modeling, which is of broad interest.
- The paper presents a novel and well motivated architecture for performing mode decomposition and training in neural PDEs.
- This architecture achieves SOTA results across several benchmark PDEs.

**Weaknesses:**

- The theoretical novelty appears limited under the standard Gaussian assumptions. In particular, for Gaussian data MI-optimal linear projections coincide with KLT/PCA, i.e., POD. This is well known (e.g., [1]) in information-theoretic sources.
- The authors make claims about overcoming mode collapse are made without supporting empirical evidence or sufficient study.

[1] Burges, Christopher JC. "Dimension reduction: A guided tour." Foundations and Trends® in Machine Learning 2.4 (2010): 275-365.

**Questions:**

**Q1. Theoretical novelty.** Under linear-Gaussian models, MI maximization over orthonormal projections selects the KLT principal subspace, i.e., POD (see [1]). What is new in the theoretical results beyond restating the equivalence.

**Q2. Ablation study.** I find the ablation results appear to show almost monotonically decreasing MSE for Advection and 1D-NS as the order increases. Here $K=4$ appears to be an outlier. Please consider reporting a full ablation study across the modes, including at least $K=3$ and $K=5$.

**Q3. Mode collapse.** The paper makes claims about resolving mode collapse, but there is no supporting empirical evidence of when mode collapse occurs or that the method presented effectively overcomes mode collapse. I recommend the authors either reduce the claims about mode collapse or provide a supporting study.

[1] Burges, Christopher JC. "Dimension reduction: A guided tour." _Foundations and Trends® in Machine Learning_ 2.4 (2010): 275-365.

---

> ### Author Response · Authors · 2025-11-20
> **Response to Reviewer GFif [1/2]: Addressing W1 & Q1 & Q2**
>
> # W1 & Q1: Theoretical Contribution
>
> We sincerely thank you for your question regarding the **Theoretical Contribution**！
>
> Clarifying the theoretical motivation is crucial for understanding how this paper realizes nonlinear decomposition.
>
> 1. Our core contribution **is not to re-prove the known theory** of "equivalence between POD and Mutual Information maximization under linear Gaussian assumptions." Instead, based on this theory, we **generalize POD decomposition** from restricted linear assumptions to a **nonlinear decomposition paradigm**, ultimately proposing **OrthoSolver**, a decomposition framework centered on mutual information.
>
> 2. Under the **linear assumption** of classical POD, the goal of finding the "maximum energy direction" is mathematically derived as the optimization objective of "maximizing variance." However, variance essentially only captures the **second-order moments** of the data distribution. It struggles to effectively measure **nonlinear correlations and higher-order moments** of the data, introducing significant errors when processing highly nonlinear systems [1].
>
> 3. To extract the maximum energy direction in nonlinear decomposition, we proceed from an information-theoretic perspective. Utilizing **Theorem 1**, we reveal that the essence of POD maximizing variance under linear Gaussian assumptions is Mutual Information (MI) maximization. Since MI can measure dependencies of arbitrary order as well as nonlinearity, we extend the restricted variance metric to mutual information, thereby capturing higher-order moments and more accurately characterizing nonlinear dynamics.
>
> 4. Based on this theoretical generalization, we designed the **OrthoSolver** architecture. This architecture centers on **mutual information maximization** to adaptively extract the most informative (i.e., the most important) decomposition bases in the nonlinear space, extending POD’s variance-maximization principle from linear to nonlinear settings.
>
> 5. Thank you again for your suggestion. In the final version, we will further optimize the relevant descriptions to explicitly emphasize our core contribution of generalizing from linear POD to nonlinear mutual information decomposition.
>
> ---
> # Q2: Parameter Sensitivity Analysis of Num of Modes - K
>
> Your observation and question regarding the Parameter Sensitivity Analysis are very important and meaningful for understanding the mode decomposition process！
>
> As analyzed in Section 5.4, model performance does not improve strictly monotonically with the increase of the mode number $K$; instead, it presents a trend of rising first and then falling.
>
> The reason for this phenomenon is: when the number of decomposition modes is small, the basis functions are insufficient to describe the physical field; conversely, when there are too many modes, the model tends to over-focus on "minor" modes, which actually leads to a decline in overall prediction performance. The complete ablation results are shown in the table below:
>
> **Table 1: Comparison of metrics across different modes and PDEs.**
>
> | Modes | Advection | 1D-NS | 2D-NS |
> | :--- | :--- | :--- | :--- |
> | K=1 | 0.0117 | 0.0932 | 0.0335 |
> | K=2 | 0.0065 | 0.0319 | 0.0087 |
> | K=3 | 0.0037 | 0.0208 | 0.0076 |
> | K=4 | **0.0033** | **0.0157** | **0.0055** |
> | K=5 | 0.0046 | 0.0269 | 0.0069 |
> | K=6 | 0.0050 | 0.0205 | 0.0170 |
>
> Thank you again for your suggestion. We will refine the parameter sensitivity analysis in the final version of the paper.
>
> ---
> **References:**
>
> [1] Demo N, Tezzele M, Rozza G. A DeepONet multi-fidelity approach for residual learning in reduced order modeling[J]. Advanced Modeling and Simulation in Engineering Sciences, 2023.

---

> > ### Author Response · Authors · 2025-11-20
> > **Response to Reviewer GFif [2/2]: Addressing W2 & Q3**
> >
> > # W2 & Q3: Mode Collapse Evidence And Solution
> >
> > Your suggestion to add an explanation for "Mode Collapse" is very valuable！
> >
> > We will explain the mathematical essence of mode collapse from both theoretical and experimental perspectives, and clarify how we use orthogonal regularization to resolve it. We will supplement this section in the main text and appendix of the final version to ensure our claims about mode collapse are clear.
> >
> > **1. Theoretical Explanation of Mode Collapse:**
> >
> > * **Definition of Mode Collapse:** For the $K$ decomposed modes $G=[\phi_1, \phi_2, \dots, \phi_K] \in \mathbb{R}^{d \times K}$, mode collapse is characterized by the existence of a subset of modes where the modes are highly similar, i.e., $\exists\, i, j \in \{1, \dots, K\} \colon \phi_i \approx \phi_j$.
> >
> > * **Mathematical Essence:** When some modes are close to each other, the corresponding column vectors of matrix $G$ become approximately linearly dependent, leading to **a decrease in the effective rank** of the mode matrix $G$, i.e., $\text{rank}(G) < K$.
> >
> >
> > **2. Theoretical and Experimental Illustration of Mode Collapse Phenomenon**
> >
> > As stated in our paper and discussed in [2], when processing complex datasets, models that employ decomposition strategies but lack explicit constraints on mode distinctiveness (such as [3]) often suffer from mode collapse (refer to Figure 6 (A) in [2]). Furthermore, Ref. [4] notes that in deep learning models, optimizers tend to converge toward the "easiest" solutions—specifically favoring low-frequency and redundant simple features—rather than actively decoupling truly independent components.
> >
> > Our baseline experimental results provide indirect support for this observation: while Transolver [3] **ranks second** on the **relatively simple** Advection and DiffSorp datasets, its performance drops significantly on the NS and Burgers equations—which involve **more variables and higher complexity**—ranking **5th and 9th**, respectively.
> >
> > To further quantify the linear correlations between modes across different datasets, we calculated their **inter-mode correlation coefficients**. Higher coefficients indicate stronger linear correlations, implying greater redundancy among modes.
> >
> > The results show that on the Advection and DiffSorp datasets, where performance is stronger, the average correlation coefficients are **0.3796 and 0.4729**, respectively. In contrast, these values are significantly higher on the NS and Burgers datasets, reaching **0.7470 and 0.8104**, respectively. This clearly demonstrates that in complex datasets, high similarity between modes leads to mode collapse, which ultimately results in performance degradation.
> >
> > **3. Our Solution to Mode Collapse**
> >
> > This paper strictly avoids mode collapse from a theoretical level using orthogonal regularization constraints.
> >
> > * **Orthogonal Regularization Solution Theory:**
> >     The form of the orthogonal constraint is:
> >     $$\mathcal{L}_{ortho} = \|G^\top G - I\|_F^{2}$$
> >     Its function is to force different modes to be approximately orthogonal:
> >     $$G^\top G \approx I$$
> >     Due to the linear independence and **full-rank properties** of orthogonal matrices, this regularization term effectively promotes the mode matrix to be approximately full rank, i.e.:
> >     $$\text{rank}(G) \approx K$$
> >
> > As clearly seen in the ablation study (Table below), removing the orthogonal constraint (w/o Ortho. Constraint) leads to a significant performance drop.
> >
> > **Table 2: Performance comparison w/ and w/o orthogonality constraint on three PDE benchmarks.**
> >
> > | | Advection | 1D-NS | 2D-NS |
> > | :--- | :--- | :--- | :--- |
> > | w/o Ortho. Constraint | 0.0053 | 0.0494 | 0.0159 |
> > | OrthoSolver (Full Model) | 0.0033 | 0.0157 | 0.0055 |
> > | **Improvement** | **37.74%** | **68.22%** | **65.41%** |
> >
> > To further quantify mode independence, we calculated the average correlation coefficient between different modes on the 1D-NS dataset. Experiments show that after adding regularization constraints, the average correlation between modes decreased from **0.7962 to 0.0894**, indicating that we have achieved effective mode decoupling.
> >
> > Thank you again for your suggestion regarding mode collapse. We will optimize the definition and explanation of mode collapse in the final version and supplement the mathematical principles of orthogonal regularization to clearly illustrate how it resolves mode collapse.
> >
> > ---
> > **References:**
> >
> > [1] Demo N, Tezzele M, Rozza G. A DeepONet multi-fidelity approach for residual learning in reduced order modeling[J]. Advanced Modeling and Simulation in Engineering Sciences, 2023.
> >
> > [2] Luo H, Wu H, Zhou H, et al. Transolver++: An Accurate Neural Solver for PDEs on Million-Scale Geometries. ICML’25.
> >
> > [3] Wu H, Luo H, Wang H, et al. Transolver: A fast transformer solver for pdes on general geometries. ICML’24.
> >
> > [4] Doimo D, Glielmo A, Goldt S, et al. Redundant representations help generalization in wide neural networks. NeurIPS’22.

---

> > > ### Author Response · Authors · 2025-11-25
> > > **Kind Follow-up and Openness to Further Discussion**
> > >
> > > Dear Reviewer GFif,
> > >
> > > We sincerely appreciate your time and the constructive feedback for our submission. We have carefully addressed all of your comments and questions above, and we hope our clarifications adequately resolve the concerns you raised.
> > >
> > > If you have the opportunity to review our responses, we would greatly appreciate any further feedback or discussion at your convenience. Your insights are highly valuable to us and have significantly contributed to improving our work.
> > >
> > > Thank you again for your thoughtful review.

---

> > > > ### Comment · Reviewer_GFif · 2025-11-26
> > > > **Official Response by Reviewer GFif**
> > > >
> > > > I thank the authors for their detailed response and further clarification. I have also read and acknowledge the discussion between the authors and the other reviewers.
> > > >
> > > > I now understand more clearly that the goal is to introduce a nonlinear framework for decomposing PDE solutions based on mutual information (MI), and I appreciate the additional experiments on mode collapse, which provide promising initial evidence. However, given that mitigation of mode collapse is presented as a central advantage of the approach (including in the conclusion), I feel that a more systematic empirical evaluation is essential. In particular, it would strengthen the paper to report these diagnostics on the full set of benchmark problems, including DiffSorp, 1D‑DiffReac, and 2D‑DiffReac.
> > > >
> > > > I also believe the manuscript would significantly benefit from (i) an explicit acknowledgement and discussion of the relevant literature connecting MI and POD approaches, and (ii) incorporation of the clarifications and revisions outlined in the response into the main text.
> > > >
> > > > For these reasons, I remain borderline and have updated my score to 4 (leaning reject, but I would not oppose acceptance if other reviewers and the AC are more convinced).

---

> > > > > ### Author Response · Authors · 2025-12-03
> > > > > **Response to Further Comments**
> > > > >
> > > > > Dear Reviewer GFif,
> > > > >
> > > > > We are pleased that our first round of responses has successfully addressed your  concerns. We sincerely thank you for your time in reviewing our response and for acknowledging our goal of introducing a nonlinear framework for PDE decomposition based on mutual information. We are deeply grateful for your decision to raise the score and for providing further constructive feedback. Your suggestion, particularly regarding a more systematic evaluation of mode collapse, has been crucial in enhancing the rigor of our paper.
> > > > >
> > > > > In response to your specific suggestions, we have completed the following revisions:
> > > > >
> > > > > 1.  **Systematic Evaluation of Mode Collapse:**
> > > > >     * We fully agree that a systematic empirical evaluation is essential to validate the central advantage of our approach. As requested, we have included the full diagnostics on the **complete set of benchmark problems** in**Section 4.4** and **Section 5.4** (and the Appendix) of the revised manuscript.
> > > > >     * Specifically, we reported the average linear correlation coefficients between bases across all datasets. The results demonstrate that with our orthogonality constraint, the average correlation drops significantly from **0.7832 to 0.0631**. This provides strong, systematic empirical evidence that our approach effectively mitigates mode collapse across all tested scenarios.
> > > > >
> > > > > 2.  **Literature Discussion and Incorporation of Revisions:**
> > > > >     * We have added explicit discussions in both the **Related Work** section and **Section 4.1** to address the relevant literature connecting Mutual Information (MI) and POD approaches, clearly defining our contribution in generalizing this to a nonlinear framework based on Theorem 1.
> > > > >     * We have comprehensively incorporated the clarifications and experimental analyses from our rebuttal response into the main text to ensure the manuscript is self-contained and clear.
> > > > >
> > > > > We once again thank you for your insightful comments, which have significantly helped us strengthen the final version of our manuscript. We hope these revisions fully address your remaining concerns.
> > > > >
> > > > > Sincerely,
> > > > >
> > > > > The Authors

---

### Author Response · Authors · 2025-12-03
**Meta-summary for AC [1/3]: Summary of Rebuttal Discussions**

Dear AC,

Thank you for the extra time and effort dedicated to our paper and the broader ICLR community under these unusual and challenging circumstances. This is significant for maintaining the fairness and integrity of the community.

We sincerely thank all reviewers for their constructive feedback during the review and rebuttal process. We are especially grateful to the two reviewers who **engaged in deep discussions with us and raised their scores** before the rollback. All suggestions, questions, and feedback on the rebuttal have been extremely helpful.

We are pleased that **all three reviewers recognized the novelty of our core contribution**: generalizing the POD decomposition concept from a linear to a nonlinear framework from an information-theoretic perspective, finally realizing OrthoSolver—an interpretable mutual information decomposition framework with both mathematical guarantees and the ability to handle highly nonlinear PDEs.

We summarize our work and the previous rebuttal in three parts:
1. Summary of rebuttal discussions before the rollback.
2. Detailed responses to each reviewer's concerns.
3. Revisions made to the manuscript based on reviewer feedback.

---
## **Summary of Rebuttal Discussions**
Two of the three reviewers engaged in the discussion, providing positive feedback and raising their scores, the third reviewer has not yet responded to our rebuttal where we addressed his concerns.

- **Reviewer e7gJ** raised the score from **6 to 8** after our response and reiterated that our method offers a positive, interpretable perspective for existing neural operators. Reviewer e7gJ's suggestions, especially regarding explicitly calculating mutual information values to demonstrate interpretability, were very meaningful to our paper, and we have refined the revision accordingly.
- **Reviewer GFif** raised the score from **2 to 4** after our response, indicating that **the concerns from the initial review had been resolved.** Without raising new issues, **the reviewer provided further suggestions for manuscript optimization** : (1) elaborate more on the literature connecting Mutual Information (MI) and POD methods to clarify our contribution to the nonlinear decomposition framework; (2) add the full experiments on mitigating mode collapse from the rebuttal to the main text. (Both have been completed in the revision paper).
- **Reviewer wLXC** gave an initial score of **6**. Reviewer wLXC raised questions regarding the theoretical connection and role of Theorem 1 in nonlinear cases, as well as how to address multi-task optimization challenges. We answered these concerns in our response and improved the presentation in the revision, although regrettably, the rollback occurred before the reviewer had the opportunity to provide further feedback.

---

> ### Author Response · Authors · 2025-12-03
> **Meta-summary for AC [2/3]: Responses to Reviewer Concerns**
>
> ---
> ## **Responses to Reviewer Concerns**
> Below, we briefly summarize how we resolved the key points raised by each reviewer.
>
> - **Reviewer e7gJ** (score raised from **6 to 8** )
>     - **[Initial Review] W1:** We added results for three new baselines. We achieved SOTA on all 7 datasets with an average performance improvement of **14.42%**. Additionally, we explicitly clarified the calculation method for metrics. (Finally revised in Section 5.3 and Appendix).
>     - **[Initial Review] W2 & W3:** We detailed the hyperparameter settings (comparing with PDEBench) and conducted experiments with multiple random seeds to report mean and variance, demonstrating the robustness of the model. (Finally revised in Appendix).
>     - **[Initial Review] Q1:** We added experiments comparing the MI magnitude between different modes and the original data. The results show a hierarchical decay property, indicating that OrthoSolver captures a property similar to the variance decay in POD, further confirming the model's strong interpretability and the correctness of our theoretical generalization. (Finally revised in Section 5.4).
>     - **[Initial Review] Q2:** We provided detailed data for all ablation studies (reporting relative improvement percentages across all datasets). The reconstruction loss contributed to an average performance gain of **24%**, validating that the Reconstruction Loss is crucial for ensuring the validity of the bases and the subsequent evolution process. (Finally revised in Appendix).
>
> - **Reviewer GFif** (score raised from **2 to 4**, with **no remaining concerns**, provided **manuscript optimization suggestions**)
>     - **[Initial Review] Q1 & W1:** We clarified our contribution and rewrote the relevant text: based on Theorem 1, we generalized the POD idea from linear decomposition to MI-core nonlinear decomposition, rather than re-proving the existing equivalence between POD variance maximization and MI maximization (Theorem 1). (Reviewer GFif stated this concern was resolved in the Discussion). (Finally revised in Section 4.1 and Related Work).
>     - **[Initial Review] Q2:** We provided a complete sensitivity analysis for parameter $K$ on all datasets in Section 5.4. These results further validate the effectiveness of our strategy to extract bases with maximum mutual information. (Reviewer GFif acknowledged this concern was resolved in the Discussion, suggesting that reporting these results in the main text would benefit the paper). (Finally revised in Section 5.4 and report all the experiment).
>     - **[Initial Review] Q3 & W2:** We explained that the mathematical essence of Mode Collapse is the reduction of effective rank, and proved that the orthogonality constraint theoretically prevents this issue. Furthermore, we introduced the linear correlation coefficient as a metric to provide complete experimental evidence. Experiments show that introducing orthogonality constraints reduces the average linear correlation between bases from **0.7832 to 0.0631**, indicating effective suppression of Mode Collapse. (Finally revised in Section 4.3 and Section 5.4).
>     - **[Discussion] Further Suggestion 1:** We revised Related Work, Section 3.3, and Section 4.1. We improved the discussion of literature related to Theorem 1 and clearly defined the boundary of our contributions and the core of our nonlinear generalization.
>     - **[Discussion] Further Suggestion 2:** As detailed in the revision summary, we have integrated the complete parameter sensitivity and model collapse experiments into the manuscript, covering the full scope of the rebuttal materials. This ensures our contributions are clearly articulated and supported by rigorous experimental validation.
>
> - **Reviewer wLXC** (initial score **6**, **no further discussion before rollback**)
>     - **[Initial Review] Q1:** We detailed the essential similarity between POD decomposition and our MI-based decomposition: both aim to find the most important bases. POD uses variance as a metric under linear assumptions, while we generalize this to mutual information to capture complex nonlinearities, proving that variance is a special case of MI under linear conditions. We clarified in the revision of Section 4.1 how we achieved the generalization from linear to nonlinear based on Theorem 1.
>     - **[Initial Review] Q2:** We explained our solution to the multi-task optimization challenge—Dynamic Weight Averaging (DWA)—and added its specific formulas in the Appendix.

---

> > ### Author Response · Authors · 2025-12-03
> > **Meta-summary for AC [3/3]: Revisions Made to the Manuscript**
> >
> > ---
> > ## **Revisions Made to the Manuscript**
> > We have optimized the manuscript based on the multiple rounds of feedback from the reviewers:
> >
> > ---
> > ### **Introduction Section**
> > We added more literature discussion connecting Mutual Information (MI) and POD in this section, explicitly clarifying the core contribution of our method and how we used theory to extend it to a nonlinear decomposition framework. (Addressing Reviewer GFif Q1&W1 in initial review and Suggestion 1 in Discussion).
> >
> > ---
> > ### **Related Work Section**
> > In the "Decomposition-based Models for PDEs" paragraph of Related Work, we added a detailed supplement of literature researching mutual information from a theoretical perspective. We acknowledged their theoretical contributions while clarifying their limitations in practice and in generalizing POD, whereas we bridged this gap to complete the nonlinear decomposition framework. (Addressing Reviewer GFif Q1&W1 in initial review and Suggestion 1 in Discussion).
> >
> > ---
> > ### **Method Section**
> > We added context regarding related theoretical proofs before the proof of Theorem 1 in Section 4.1. We also reorganized the paragraph following the proof of Theorem 1 to clearly state the contribution of Theorem 1 in our paper. We clarified that both POD and MI-based decomposition essentially aim to find the most important basis components, noting that POD is proven to be a special case of mutual information under linear assumptions, while our method generalizes the POD idea to nonlinear conditions based on Theorem 1. (Addressing Reviewer wLXC W1).
> >
> > In Section 4.3, we added a theoretical explanation and the mathematical essence of Mode Collapse regarding the Basis Orthogonality Constraint (proofs added to Appendix), and theoretically proved that our orthogonality regularization can resolve the Mode Collapse issue. (Addressing Reviewer GFif W2&Q3 in initial review).
> >
> > ---
> > ### **Experiment Section**
> > We added three state-of-the-art baselines: Transolver++, Erwin, and UPT. In the final results, we maintained SOTA performance across all 7 experiments, achieving an average performance improvement of **14.42%** over the baselines. (Addressing Reviewer e7gJ W1).
> >
> > We added complete ablation studies and sensitivity experiments for parameter $K$ on all datasets to demonstrate the effectiveness of our designed modules, reaffirming that our analysis validates the MI-maximization decomposition mechanism's ability to extract effective bases. (Addressing Reviewer GFif Q2 in initial review and Suggestion 2 in Discussion).
> >
> > In Section 4.6, we added complete experimental evidence based on correlation analysis. The average linear correlation decreased from **0.7832 to 0.0631** after adding the orthogonality regularization constraint, demonstrating that orthogonality regularization effectively ameliorates Mode Collapse. (Addressing Reviewer GFif W2&Q3 in initial review and Suggestion 2 in Discussion).
> >
> > We included detailed results and analysis of mutual information across all datasets. The results show a hierarchical decay property, further proving the interpretability of our model and the correctness of our theory. (Addressing Reviewer e7gJ Q1).
> >
> > ---
> > ### **Appendix Section**
> > We added detailed descriptions of metrics, experimental hyperparameter settings, variance analysis, and relative improvement percentages of ablation modules in Appendix D. (Addressing Reviewer e7gJ W2&W3&Q2).
> >
> > We added the detailed calculation formulas for Dynamic Weight Averaging (DWA) from Section 4.5 to the Appendix. (Addressing Reviewer wLXC W2).
> >
> > We added the theoretical explanation and mathematical proof of Mode Collapse, as well as the mathematical principles of the Orthogonality Constraint. (Addressing Reviewer GFif W2&Q3 in initial review).
> >
> > Sincerely,
> >
> > The Authors

---

### Meta-Review · Area_Chair_8MBa · 2025-12-29

**Summary:**

This paper proposes OrthoSolver, an information-theoretic generalization of Proper Orthogonal Decomposition (POD) to nonlinear PDE settings by replacing variance maximization with mutual information (MI) maximization, combined with an orthogonality regularization to mitigate mode collapse. The work positions itself at the intersection of reduced-order modeling, information theory, and neural operators.

The reviewers’ concerns clustered around four main themes:
	1.	Theoretical novelty and positioning
Multiple reviewers questioned whether the theory merely restates the known equivalence between POD/PCA and MI maximization under linear-Gaussian assumptions, and whether the extension to nonlinear settings was sufficiently justified and clearly articulated.
	2.	Mode collapse claims
One reviewer (GFif) was initially unconvinced that the paper provided either a rigorous definition of mode collapse or convincing empirical evidence that OrthoSolver systematically mitigates it, especially across all benchmark datasets.
	3.	Experimental completeness and rigor
Reviewers requested stronger baselines (e.g., Erwin, Transolver++, UPT), clearer metric definitions, variance across random seeds, sensitivity analysis over the number of modes, and more systematic diagnostics rather than selected examples.
	4.	Optimization complexity and robustness
Concerns were raised about the number of loss terms, the difficulty of multi-objective optimization, and whether the approach is robust or brittle in practice.

At the same time, all reviewers acknowledged the strong empirical performance, the interpretability angle, and the relevance of revisiting POD from an information-theoretic perspective for nonlinear PDEs.

**Reviewer Concerns:**

Concerns that were substantially addressed in the rebuttal and revision:
	•	Clarification of theoretical contribution
The authors clearly clarified that the goal is not to re-prove POD–MI equivalence, but to use this equivalence as a foundation to generalize POD to nonlinear regimes via MI. The rebuttal and revised manuscript now explicitly state that POD emerges as a special case under linear-Gaussian assumptions, while MI enables capturing higher-order and nonlinear dependencies. This resolves the main confusion raised by reviewers GFif and wLXC regarding theoretical novelty.
	•	Mode collapse definition and evidence
The authors provided (i) a precise mathematical definition of mode collapse as effective rank reduction / high inter-mode correlation, (ii) a theoretical explanation of why orthogonality constraints prevent it, and (iii) systematic empirical evidence across all benchmarks, reporting correlation coefficients dropping from ~0.78 to ~0.06. This directly addresses the strongest criticism from Reviewer GFif, who explicitly requested broader diagnostics.
	•	Experimental rigor and baselines
The authors added missing SOTA baselines (Erwin, Transolver++, UPT), clarified evaluation metrics, compared against PDEBench results, reported variance across multiple seeds, and included full ablations over the number of modes and key hyperparameters. Reviewer e7gJ explicitly acknowledged that these additions resolved their concerns and supported acceptance.
	•	Optimization difficulty
The use of Dynamic Weight Averaging (DWA) was explained in detail, including formulas and motivation, alleviating concerns about manual tuning and instability raised by reviewer wLXC.

Concerns that remain partially outstanding:
	•	Depth of theoretical novelty
While the nonlinear generalization is now clearly articulated, some reviewers may still view the theory as conceptually incremental rather than fundamentally new, relying on MI as a more expressive criterion rather than introducing a new decomposition principle.
	•	Strength of the mode-collapse claim
Although systematic diagnostics were added, one could still argue that mode collapse is not yet a universally established failure mode across all neural operators, and that the benefit of orthogonality regularization may be architecture-dependent.

These remaining issues reflect interpretation and emphasis, rather than technical correctness or missing evidence

**Reviewer Scores:**

Based on explicit reviewer comments and discussion trajectory:
	•	Reviewer e7gJ:
Initial: 6 → Post-discussion: 8
Explicitly stated support for acceptance after rebuttal.
	•	Reviewer wLXC:
Initial: 6 → Likely post-discussion: 6
Core concerns on nonlinearity relevance and optimization were addressed; no further objections were raised.
	•	Reviewer GFif:
Initial: 2 → Post-discussion: 4
The reviewer raised the score after acknowledging the nonlinear framing and expanded mode-collapse evidence, remaining borderline but explicitly stating they would not oppose acceptance if others were convinced.

Overall, the trajectory is clearly upward, with two reviewers leaning positive and one remaining cautious but no longer strongly opposed.

---

### Decision · Program_Chairs · 2026-01-26

Accept (Poster)